# Nsun4 and Mettl3 mediated translational reprogramming of Sox9 promotes BMSC chondrogenic differentiation

Lin Yang [1,10], Zhenxing Ren[2,10], Shenyu Yan[3,10], Ling Zhao[4], Jie Liu[1], Lijun Zhao[1], Zhen Li[1], Shanyu Ye[5], Aijun Liu[5], Xichan Li[6], Jiasong Guo [7], Wei Zhao [8], Weihong Kuang[9], Helu Liu[1✉] & Dongfeng Chen [5✉]

The chondrogenic differentiation of bone marrow-derived mesenchymal stem cells (BMSCs) has been used in the treatment and repair of cartilage defects; however, the in-depth regulatory mechanisms by which RNA modifications are involved in this process are still poorly understood. Here, we found that Sox9, a critical transcription factor that mediates chondrogenic differentiation, exhibited enhanced translation by ribosome sequencing in chondrogenic pellets, which was accompanied by increased 5-methylcytosine (m5C) and N6-methyladenosine (m6A) levels. Nsun4-mediated m5C and Mettl3-mediated m6A modifications were required for Sox9-regulated chondrogenic differentiation. Interestingly, we showed that in the 3′UTR of *Sox9* mRNA, Nsun4 catalyzed the m5C modification and Mettl3 catalyzed the m6A modification. Furthermore, we found that Nsun4 and Mettl3 co-regulated the translational reprogramming of Sox9 via the formation of a complex. Surface plasmon resonance (SPR) assays showed that this complex was assembled along with the recruitment of Ythdf2 and eEF1α-1. Moreover, BMSCs overexpressing Mettl3 and Nsun4 can promote the repair of cartilage defects in vivo. Taken together, our study demonstrates that m5C and m6A co-regulate the translation of Sox9 during the chondrogenic differentiation of BMSCs, which provides a therapeutic target for clinical implications.

[1] Shenzhen Hospital of Integrated Traditional Chinese and Western Medicine, Shenzhen 518101 Guangdong, China. [2] Shanghai Jiao Tong University Affiliated Sixth People's Hospital, Shanghai 200233, China. [3] Department of Pharmacology and Pharmacy, LKS Faculty of Medicine, The University of Hong Kong, Hong Kong 61001-89999, China. [4] Academy of Integrative Medicine, Shanghai University of Traditional Chinese Medicine, Shanghai 201203, China. [5] Department of Anatomy, School of Basic Medical Sciences, Guangzhou University of Chinese Medicine, Guangzhou 510006, China. [6] School of Chinese Herbal Medicine, Guangzhou University of Chinese Medicine, Guangzhou 510006, China. [7] Department of Histology and Embryology, Southern Medical University, Guangzhou 510515, China. [8] RNA Biomedical Institute, Sun Yat-sen Memorial Hospital, Sun Yat-sen University, Guangzhou 510120, China. [9] Guangdong Key Laboratory for Research and Development of Natural Drugs, Key Laboratory of Research and Development of New Medical Materials of Guangdong Medical University, School of Pharmacy, Guangdong Medical University, Dongguan, China. [10] These authors contributed equally: Lin Yang, Zhenxing Ren, Shenyu Yan. ✉email: liuhelu@21cn.com; chen888@gzucm.edu.cn

Articular cartilage defects of the knee are prevalent, 36% of athletes have full-thickness cartilage defects[1]. Osteoarthritis (OA) mainly characterized by cartilage damage is a degenerative joint disease with a high prevalence in the elderly population[2]. Chondrocytes are the resident cell type in articular cartilage and they have poor spontaneous healing ability. Although syndrome-based therapies and cartilage regeneration methods have been developed, current technologies for recovering cartilage defects are still limited[3,4]. Stem-cell-induced tissue regeneration has emerged as a promising strategy for cartilage regeneration and repair[5–7].

Mesenchymal stem cells (MSCs) are a reliable resource for tissue regeneration[8]. With the advantages of high proliferation, high plasticity and multipotency, MSC-based alternative repair strategies have been widely applied in the treatment of OA, osteoporosis, and autoimmune diseases[9–11]. MSCs can be isolated from multiple tissues (e.g. the umbilical cord, umbilical cord blood, bone marrow, adipose tissue, etc.). Among them, bone marrow-derived MSCs (BMSCs) with a high proliferative capability are the most frequently utilized in studies of chondrogenic differentiation[12,13].

SRY-related high-mobility group box 9 (Sox9) is responsible for controlling the expression of chondrocyte proteins, such as collagen type II, type IX, type XI and aggrecan[14,15]. In osteoarthritic chondrocytes, the transcriptional activity of Sox9 was found to be attenuated, which resulted in a downregulation of the matrix anabolism of articular cartilage[16]. Inactivation of Sox9 results in the complete absence of mesenchymal condensation, ultimately leading to abnormalities in cartilage formation[17,18]. In addition, overexpression of Sox9 promoted the chondrogenic differentiation of BMSCs[19], and administration of the Sox9 vector delayed premature hypertrophic and osteogenic differentiation in the constructs[20]. The key role of Sox9 in the treatment of cartilage diseases has been frequently reported; however, the mechanisms by which RNA modifications regulate Sox9 are less understood.

RNA modifications are key cellular processes. To date, more than 170 modifications have been reported[21], of which, in eukaryotes, m5C and m6A modifications are the two predominant types that contribute to gene regulation through a variety of different mechanisms, including pre-mRNA splicing, mRNA export, translation, and mRNA degradation[22,23] m5C is catalysed by enzymes of the NOL1/NOP2/SUN domain (Nsun) family, which includes seven members (Nsun1-7)[24]. And the m6A modification is catalysed by an unidentified methyltransferase complex mainly containing at least one subunit methyltransferase like 3 (Mettl3)[25]. Recently, m5C detection approaches have been applied to RNAs from mouse embryonic stem cells, which supported the presence of m5C in mRNA. Subsequently, Nsun3 was characterized as an important regulator of stem cell fate. Nsun3 inactivation attenuated induction of mitochondrial reactive oxygen species (ROS) upon stress, which may affect gene expression programs upon differentiation. Mettl3 severely impairs embryonic stem cell differentiation and results in early embryonic lethality. Together, studies have proven that m5C and m6A play a regulatory role in stem cell differentiation[26–29]. However, the methylation mechanism by which m5C and m6A methylation plays a role in the chondrogenic differentiation of BMSCs has not yet been reported.

Given the importance of m5C and m6A regulation in stem cell differentiation, we hypothesized that these modifications might play a key role in the chondrogenic differentiation of BMSCs. In this study, by performing ribosome footprinting (Ribo-seq), we found enhanced translation of Sox9 with increased m5C and m6A levels during chondrogenic differentiation. Nsun4-mediated m5C and Mettl3-mediated m6A regulate Sox9 mRNA in the 3'UTR, which in turn regulates the translation of Sox9 by recruiting Ythdf2 and eEF1α-1. Our study revealed that comethylation-induced translation of Sox9 during chondrogenic differentiation expands the therapeutic potential for cartilage defects.

## Results

**The level of m5C and m6A modification of Sox9 is increased after chondrogenic differentiation.** Transforming growth factor beta 3 (TGF-β3) has been considered as the classical inducer of BMSC chondrogenesis[30]. We treated BMSCs from rats with TGF-β3 for 7 days to induce chondrogenic pellets (Fig. 1a). The marker genes of chondrogenic differentiation including Sox9, aggrecan, and Col2 were increased during cartilage differentiation (Fig. 1b), implying that the chondrogenesis model was successfully established.

Sox9 underlies chondrocyte differentiation by transcriptionally activating markers of overtly differentiated chondrocytes, such as aggrecan and Col2[31]. To investigate the translational mechanism of the Sox9 gene, Ribo-seq was performed. Principal component analysis (PCA) demonstrated good reproducibility among the three replicates of each group (Supplementary Fig. 2a). The ribosome distribution around genes was highly increased after chondrogenic differentiation (Fig. 1c). Furthermore, the ribosome footprint (RFs) distribution around Sox9 mRNA was highly increased at the same time (Fig. 1d).

m5C and m6A are two predominant internal modifications for mRNA fate modulation, whether they regulate the translation of Sox9 has not been studied. A dot blot study of total RNA revealed that m5C and m6A levels were significantly increased in chondrogenic pellets (Fig. 1e, f). This was further confirmed by the results obtained in LC-MS/MS analysis (Supplementary Fig. 1a, b). Specially, a ribonucleoprotein immunoprecipitation-quantitative PCR (RIP-qPCR) assay showed that the m5C and m6A levels on Sox9 mRNA were remarkably increased in the chondrogenic pellet group (Fig. 1g, h). These results indicate an increase in m5C and m6A sites on Sox9 mRNA after chondrogenic differentiation.

**Nsun4 and Mettl3 are required for Sox9-triggered chondrogenic differentiation.** It is known that m5Cs sites in RNAs are introduced by members of the NOL1/NOP2/SUN domain (Nsun) family, which contains seven members (Nsun1-7) in human[32]. We performed RNA-sequencing and RT-qPCR to measure the expressions of the Nsun genes, and the results showed that the gene expression of Nsun4 was decreased while the Nsun6 mRNA level was increased (Supplementary Fig. 2b–d). To clarify the role of Nsun4 and Nsun6 in the regulation of m5C levels, we individually downregulated Nsun4 and Nsun6 gene expression in BMSCs by using small interfering RNAs (siRNAs) (Supplementary Fig. 2e, f). The results showed that siNsun4 cells had lower level of m5C (Fig. 2h; Supplementary Fig. 2g), but siNsun6 cells showed no differences during differentiation (Supplementary Fig. 2h). These data indicate that the methylation of m5C is dependent on Nsun4 during chondrogenic differentiation. m6A is catalyzed by a complex multicomponent enzyme, of which Mettl3 is the critical S-adenosyl-L-methionine-binding subunit. To investigate the role of Mettl3 in chondrogenic differentiation, siRNA was used to knock down the expression of Mettl3 (Supplementary Fig. 2i). Consistently, the m6A level was lower in the Mettl3-deficient group than in the negative control (NC) group (Fig. 2i). These data indicate that Mettl3 regulates m6A after chondrogenic differentiation.

During chondrogenic differentiation, the protein and mRNA expression of Nsun4 was downregulated (Fig. 2a, b), while the Mettl3 protein and mRNA levels were increased after

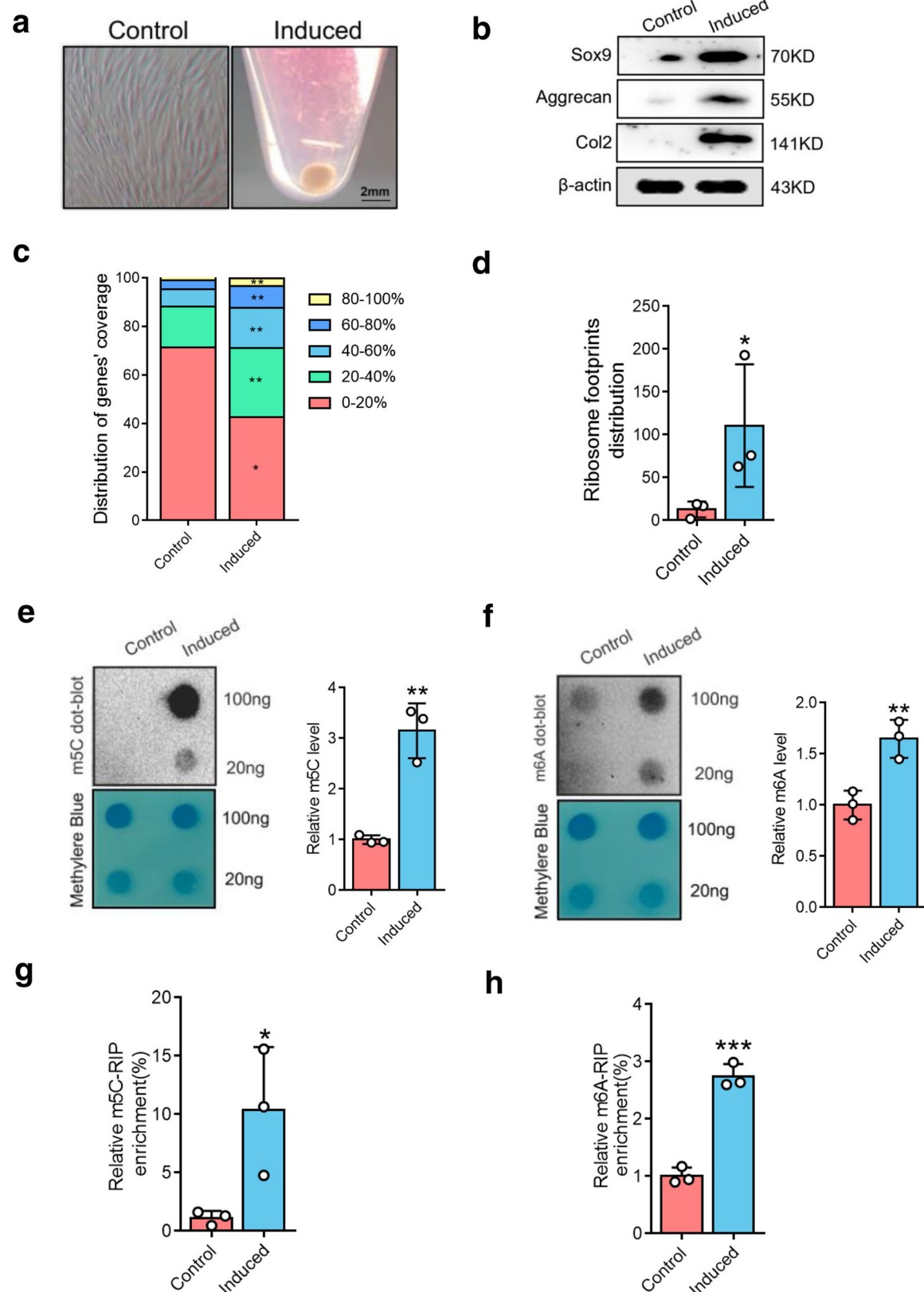

chondrogenic differentiation (Fig. 2c, d). Conversely, chondrogenic pellets could not be induced when Nsun4 and Mettl3 were silenced (Fig. 2e). Chondrogenic differentiation markers, including Sox9, Col2 and aggrecan, were also suppressed compared with those in NC cells (Fig. 2f, g). To investigate whether Nsun4 and

Mettl3 participate in *Sox9* mRNA modification, RIP-qPCR was used. The data revealed that Nsun4 and Mettl3 were remarkably enriched with *Sox9* mRNA (Fig. 2j, k). These findings suggest that Nsun4 and Mettl3 are required for m5C and m6A modification of *Sox9* mRNA after chondrogenic differentiation.

**Fig. 1 m5C and m6A regulate Sox9 during chondrogenic differentiation. a** The chondroblast was formed on chondrogenic differentiation of BMSCs cultured in chondrogenic induce medium. Scale bars indicate 2 mm. **b** The expressions of Sox9, Aggrecan, and Col2 were detected by western blot. β-actin is used as a loading control. **c** Ribosome footprint coverage of genes was detected in both the control and induced cells in Ribosome profiling (Ribo-seq). $n = 3$. The percentage refers to the proportion of each gene covered by ribosomal reads. **d** Comparison of differential ribosome footprint abundance on Sox9 mRNA between control and induced groups. **e** m5C were detected by dot blot in mRNAs of BMSCs during chondrogenic differentiation (left) and quantitatively analyzed (right). **f** m6A dot blot in mRNAs of BMSCs during chondrogenic differentiation (left) and quantification of m6A abundance (right). **g**, **h** m5C (**g**) and m6A (**h**) RIP-qPCR analysis of Sox9 mRNA after 7 days of chondrogenic induction. Data are presented as means ± SD from three independent experiment. *$P < 0.05$, **$p < 0.01$, ***$p < 0.001$, by Student's $t$ test.

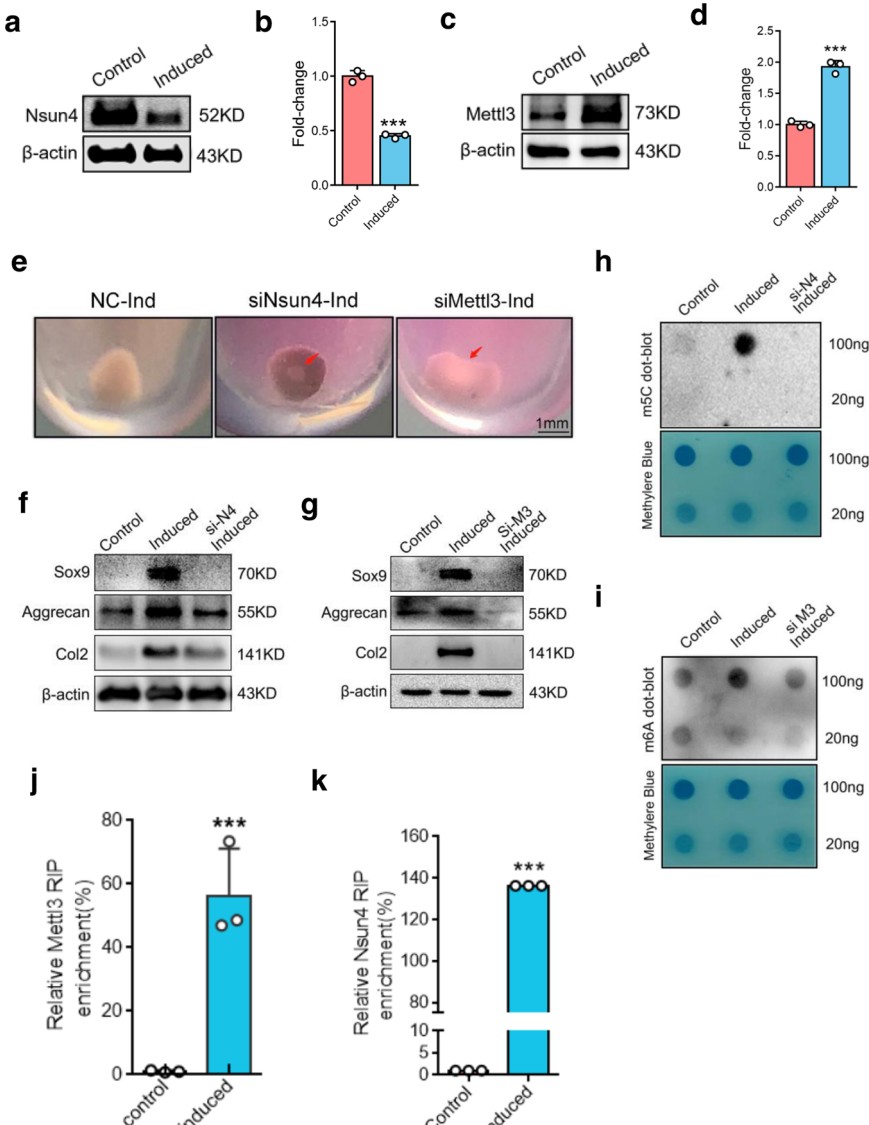

**Fig. 2 Nsun4 and Mettl3 are both required on chondrogenic differentiation. a, b** Nsun4 protein (**a**) and mRNA (**b**) expression during BMSCs chondrogenic differentiation tested by western blot and RT-qPCR. β-actin is used as a loading control. **c, d** Western blot (**c**) and RT-qPCR (**d**) showing expression of Mettl3 in control and chondrogenic induction. **e** Chondroblast formation of the Negative Control and Nsun4/Mettl3 knockdown BMSCs on chondrogenic differentiation. **f, g** Protein expressions of Sox9, Aggrecan, Col2 were measured by western blot in BMSCs transfected with siRNA of Nsun4 (**f**) and Mettl3 (**g**) on chondrogenic differentiation. **h, i** Dot blot analysis of m5C (**h**) and m6A (**i**) in total mRNA of the BMSCs transfected with or without siRNAs of Nsun4 or Mettl3 for 24 h and then further induced for 7 days. **j, k** Mettl3 (**j**) and Nsun4 (**k**) RIP-qPCR analysis of Sox9 mRNA in control or induced cells on chondrogenic differentiation. Data are presented as means ± SD from three independent experiment. ***$p < 0.001$, by Student's $t$ test.

**Nsun4 and Mettl3 co-regulate m5C and m6A methylation of *Sox9* by binding to the 3'UTR.** Given the indispensable roles of Nsun4 and Mettl3 in chondrogenic differentiation, we hypothesized that they may form a complex, and bind to a specific site on *Sox9* mRNA. First, we explored the interaction between Nsun4 and Mettl3 by performing co-immunoprecipitation (Co-IP) and

western blotting. The results showed an increased level of Mettl3 in the precipitate that was pulled down with the Nsun4 antibody compared to the control group (Fig. 3a). Additionally, an increased level of Nsun4 was shown in the precipitate pulled down with the Mettl3 antibody after chondrogenic differentiation (Fig. 3b). To determine whether Mettl3 and Nsun4 are dependent

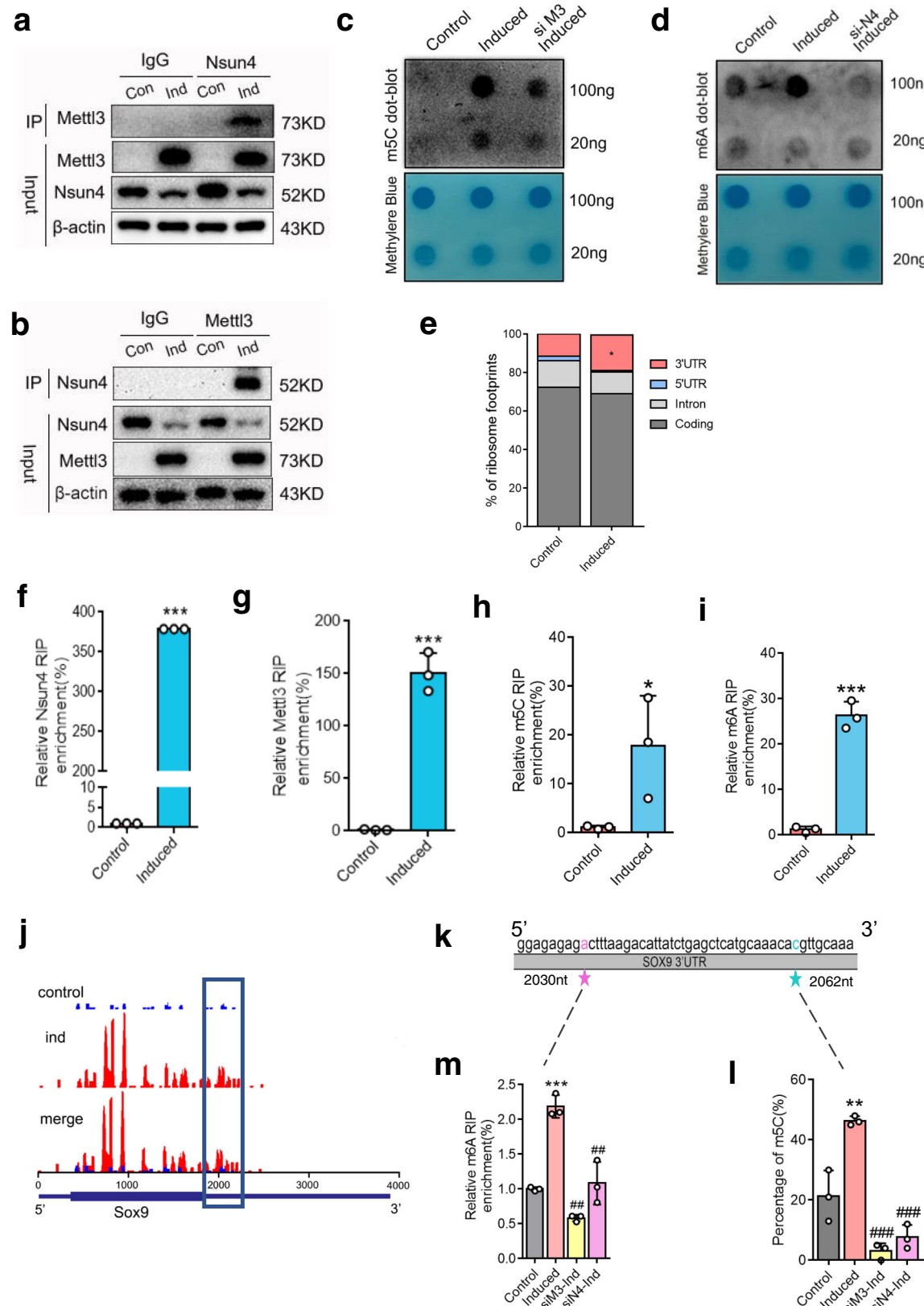

on each other, we confirmed that the expression levels of Mettl3 and Nsun4 were lower in siNsun4 and siMettl3 cells, respectively, than in the NC groups during chondrogenic differentiation (Supplementary Fig. 3a, b). Meanwhile, we individually upregulated Mettl3 and Nsun4 expression in BMSCs by AAV infection. As shown in Supplementary Fig. 3c, d, the Mettl3 and Nsun4 levels were markedly increased in Mettl3 and Nsun4-overexpressing (OE) BMSCs, respectively. We consistently observed a increase in Mettl3 and Nsun4 protein expression with Nsun4 and Mettl3 overexpression after chondrogenic differentiation (Supplementary Fig. 3e, f). Furthermore, Mettl3-silenced cells displayed a downregulated $m^5C$ level (Fig. 3c),

**Fig. 3 Nsun4 and Mettl3 co-regulate m5C and m6A methylation of Sox9 depends by binding to its 3'UTR. a** Proteins were immunoprecipitated with Nsun4 antibody from BMSCs with or without undergoing chondrogenic differentiation. Mettl3 was detected with the indicated antibody. **b** Proteins were immunoprecipitated with Mettl3 antibody from BMSCs with or without undergoing chondrogenic differentiation; Nsun4 was detected with the indicated antibody. **c, d** m5C c and m6A d dot blot of the Mettl3 or Nsun4 knockdown BMSCs undergoing chondrogenic differentiation. **e** Proportion of ribosome footprints distribution in the 5'UTR, 3'UTR, Coding or Intron region across the entire set of mRNA transcripts. **f, g** Binding of Nsun4, Mettl3 with the 3'UTR of Sox9 mRNA in control and induced cells were analyzed by Nsun4 (**f**) or Mettl3 (**g**) RIP-qPCR. **h, i** m5C (**h**) and m6A (**i**) RIP-qPCR analysis of the 3'UTR of Sox9 mRNA in control or induced cells. **j** Ribosomes were enriched in 3'UTRs of Sox9 genes from Ribo-seq data. Square marked increases of Ribosome Footprints in chondroblast cells undergoing chondrogenic differentiation. $n = 3$. **k** Schematic depiction of the m5C and m6A sites. **l** Statistical analysis for the percentage of the m5C site in the 3'UTR of the Sox9 mRNA by RNA-BisSeq. **m** BMSCs were transfected with or without siRNAs for 24 h, then cultured in chondrogenic induce medium; the m6A site in Sox9 3'UTR was analyzed by m6A RIP-qPCR. Data are presented as means ± SD from three independent experiment. Student's $t$ test were performed for (**e-i**). One way ANOVA and Tukey's multiple comparison tests were performed for (**m, l**). *$P < 0.05$, **$p < 0.01$, ***$p < 0.001$ compared with the control group; ##$p < 0.01$, ###$p < 0.001$ compared with the induced group.

and the m6A level was decreased in cells after the knockdown of Nsun4 (Fig. 3d). The m5C and m6A levels were remarkably rescued by Mettl3 overexpression, and the levels of m5C and m6A were upregulated in OE-Nsun4 cells after chondrogenic differentiation (Supplementary Fig. 3g–j). Collectively, these data suggest that Mettl3 and Nsun4 stabilize each other and form a complex that is required for regulating m5C and m6A levels after chondrogenic differentiation.

Next, we focused on the binding site on the Sox9 mRNA of the two interactive proteins interacting. Ribo-seq data showed that the distribution of RFs was upregulated in the 3'UTR region (Fig. 3e). To determine whether Nsun4 or Mettl3 participates in the methylation of the 3'UTR of Sox9 mRNA, a RIP-qPCR assay was used to investigate the interaction between the 3'UTR of Sox9 mRNA and Nsun4 or Mettl3. Our data revealed that Nsun4 and Mettl3 enrichment was significantly increased after chondrogenic differentiation (Fig. 3f, g). Moreover, by applying m5C and m6A RIP-qPCR, we verified that the m5C and m6A levels of the 3'UTR of Sox9 mRNA were significantly increased after chondrogenic differentiation (Fig. 3h, i).

The Ribo-seq data showed that ribosome peaks in 3'UTR regions ranging from the nucleotides 1950 to 2250 of Sox9 were increased after chondrogenic differentiation (Fig. 3j). Three GGAC motifs as potential m6A methylation sites were predicted in the 3'UTR of Sox9 mRNA (ranging from nucleotides 2020 to 2250) by the SRAMP prediction server (http://www.cuilab.cn/sramp) (Supplementary Fig. 5a). Meanwhile, bisulfite sequencing was used to detect the m5C in this region. To assess efficiency of bisulfite conversion treatment, we used the 28 S rRNA as positive control, as the C at position 4447 is generally 100% methylated. We performed bisulfite treatment of total RNA, followed by RT-PCR and Sanger sequencing of the C4447 encompassing region of the 28 S rRNA. The result showed a complete C-T conversion along the fragment suggesting the absence of methylation on these C residues, and no conversion of C4447 residue confirming the methylation status of this specific C residue (Supplementary Fig. 4a). To further assess the conversion rate, the RT-PCR sequencing results of 10 clones were obtained. The sequence analysis showed an average conversion rate of 99.5%, suggesting that bisulfite treatment was efficient(Supplementary Fig. 4b). The bisulfite sequencing uncovered the exact position of m5C at nucleotide 2062 after chondrogenic differentiation (Fig. 3k; Supplementary Fig. 5b).

Methylation levels of the m5C site were decreased in the Nsun4-knockdown and Mettl3-knockdown groups (Fig. 3l). Furthermore, the RNA fragments were enriched by RIP, and qPCR was used to determine that the nucleotide 2030 was the m6A methylation site near the m5C methylation site (Fig. 3k). Next, we cloned the 300-nucleotide (nt)-long WT or mutant 3'UTR truncation sequence (from 1950 to 2250) into the pmirGLO luciferase (Luc) reporter (Supplementary Fig. 5c).

Knockdown of Mettl3 decreased Luc activity and Mettl3 overexpression increased Luc activity of the WT reporter but not that of the mutant reporter, which further confirmed that nucleotide 2030 was m6A methylation site (Supplementary Fig. 5d). After deletion of Nsun4 and Mettl3, the methylation level of the m6A site was apparently downregulated (Fig. 3m). Taken together, these results indicated that Nsun4 and Mettl3 bind to the 3'UTR region of Sox9 mRNA and they modify the m5C and m6A sites.

**Nsun4 and Mettl3 regulate the translational reprogramming of Sox9 through Nsun4/Mettl3/Ythdf2/eEF1α-1 (NMYE) complex assembly.** To further explore the regulatory mechanisms of Nsun4 and Mettl3, we identified proteins that interact with Nun4. The proteins that coimmunoprecipitated with the Nsun4 antibody were visualized by silver-staining (Supplementary Fig. 6a), and immunoprecipitated proteins were gel excised, trypsin digested and analysed by liquid chromatography mass spectrometry (LC-MS). Consistently, Mettl3 was identified among the precipitated proteins. Meanwhile, ribosomal proteins and translation-related proteins were also found (Supplementary Fig. 6b). Of them, eEF1α-1, an elongation factor that regulates translation elongation in eukaryotes, was detected in the chondrogenic pellet group. Furthermore, co-immunoprecipitation (Co-IP) and western blot analysis indicated that the binding between Nsun4, Mettl3 and eEF1α-1 was increased in chondrogenic pellets compared with that in control cells, respectively (Fig. 4a, b).

m5C and m6A modifications indirectly affect the translation of RNA by recruiting specific reader proteins[33,34]. YT521-B homology domain family (Ythdf) proteins consist of three homologous members, Ythdf1, Ythdf2, and Ythdf3, which are a group of evolutionarily conserved m6A readers[35]. To explore whether the Ythdf proteins participate in the methylation of Sox9, we measured the variation in Nsun4- and Mettl3-binding Ythdf proteins after chondrogenic differentiation. Our data showed that Ythdf1, Ythdf2 and Ythdf3 could be pulled down with the Mettl3 antibody, while Ythdf2 was precipitated with the Nsun4 antibody (Fig. 4c, d), indicating that Ythdf2 specially interacted with the Nsun4-Mettl3 complex. We observed a decreased Nsun4 and Mettl3 protein levels in si-Ythdf2 cells (Fig. 4e) and the protein expression of Sox9 was decreased in si-Ythdf2 cells after chondrogenic differentiation (Fig. 4f). Furthermore, RIP-qPCR was used to investigate the interaction between the 3'UTR region of Sox9 mRNA and Ythdf2. Our data showed that Ythdf2 interaction with the 3'UTR of Sox9 mRNA was remarkably increased during chondrogenic differentiation (Fig. 4g).

The mechanisms regulating Sox9 expression through m5C and m6A co-methylation were further investigated. First, we constructed a pmirGLO-Sox9 luciferase reporter by ligatingwild type (WT) and mutant type (Mut) Sox9 3'UTR into the multiple

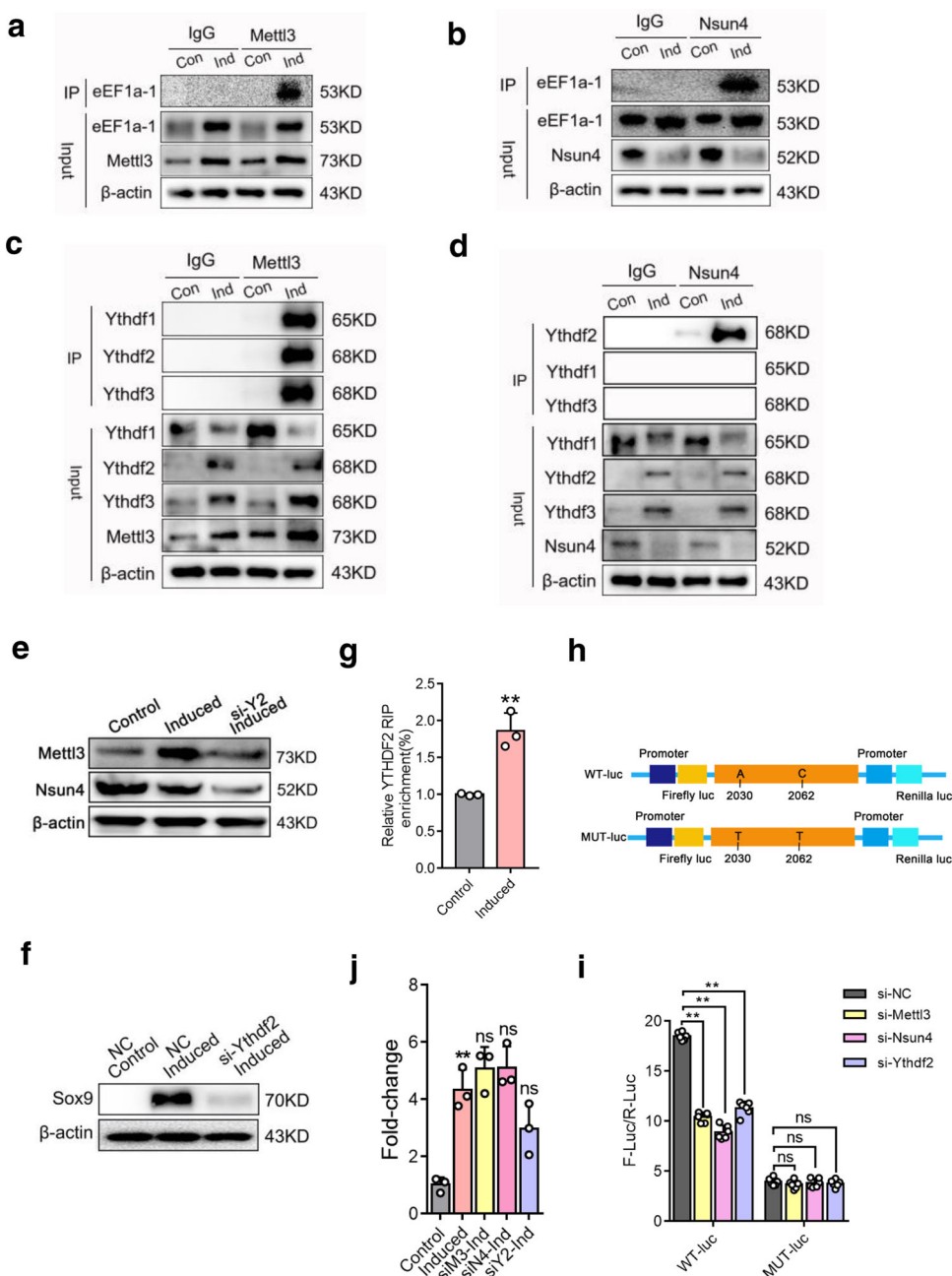

**Fig. 4 Nsun4 and Mettl3 regulate translational reprograming of Sox9 through NMYE complex assembly. a**, **b** Proteins were immunoprecipitated with Nsun4 a or Mettl3 b antibody from BMSCs with or without on chondrogenic differentiation; eEF1a-1 was detected with the indicated antibody. **c**, **d** Binding between Mettl3 c or Nsun4 d with YTHDF1-3 in control and induced cells were checked by immunoprecipitation; YTHDF2 was specifically recognized. **e**, **f** BMSCs were cultured in chondrogenic induce medium for 7 days after transfected with or without si-YTHDF2 for 24 h, the protein expression of Mettl3, Nsun4 e and Sox9 f was detected by western blot analysis. β-actin is used as a loading control. **g** YTHDF2 RIP-qPCR analysis of SOX9 mRNA in 3'UTR in the control and BMSCs undergoing chondrogenic differentiation. **h** Schematic diagram of Sox9 3'UTR wide type (WT-luc) and mutant type (MUT-luc) reporters. The 300-nt DNA sequence of the WT Sox9 3'UTR segment sequence was inserted to the dual-luciferase reporter plasmid to give rise to the WT-luc reporter. For the MUT-luc, A and C to T substitutions were made. **i** Relative dual-luciferase reporter activity of WT-luc (left) and Mut-luc (right) reporter in 293 T cells with ectopically expressed Mettl3, Nsun4 and Ythdf2 respectively. **j** BMSCs were transfected with or without siRNAs for 24 h, then cultured in chondrogenic induce medium; Sox9 mRNA was analyzed by RT-qPCR. Data are presented as means ± SD from three or six independent experiment. Student's $t$ test were performed for (**g**). One way ANOVA and Tukey's multiple comparison tests were performed for (**j**, **i**). **$p < 0.01$ compared with the control group or NC group; ns, no significant. NMYE, Nsun4, Mettl3, Ythdf2, eEF1a-1.

cloning sites (MCs) (Fig. 4h; Supplementary Fig. 5e, f). The dual-luciferase assay showed that translation efficiency of *Sox9* in the siNsun4, siMettl3 and siYthdf2 groups was significantly down-regulated compared to that in the 293 T NC group with the WT reporter (Fig. 4i left), while it was unchanged in the m$^5$C and m$^6$A site mutant reporter groups (Fig. 4i right; Supplementary Fig. 5e, f). Furthermore, the mRNA expression of *Sox9* in si-Mettl3/si-Nsun4/si-Ythdf2 cells was not different (Fig. 4j). This result indicated that the NMYE complex co-regulated the translation of *Sox9*.

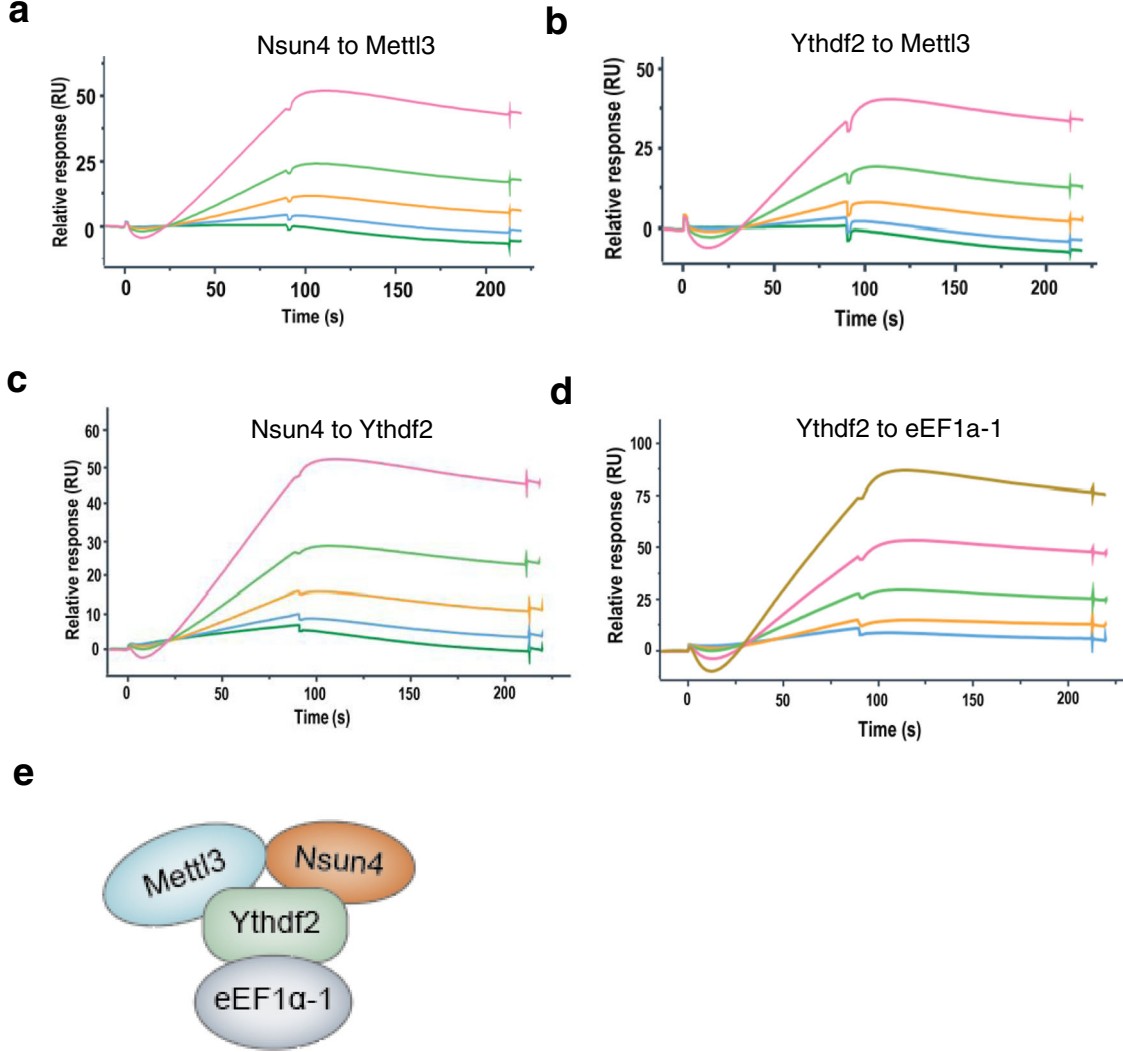

**Fig. 5 The assembly model of NMYE complex is confirmed in vitro. a**, **b** SPR sensorgram showing the binding of Mettl3 with Nsun4 (**a**) and Ythdf2 (**b**), respectively. **c**, **d** SPR sensorgram showing the binding of Ythdf2 with Nsun4 (**c**) and eEF1α-1 (**d**), respectively. Data are shown as black lines and the fitted lines in red. **e** A cartoon depicting the interaction mode of the NMYE complex.

**The assembly model of the NMYE complex was explored in vitro**. To elucidate a model of the interaction model among the NMYE complex proteins, we performed an SPR experiment to test the direct binding. His-tagged proteins were purified and the SUMO protease ULP1 was used to cleave the His-tagged protein to obtain Nsun4, Ythdf2 and eEF1α-1. The His-tagged proteins and cleaved proteins were analysed by SDS-PAGE (Supplementary Fig. 7a–d).

The His-tagged proteins were separately captured on an NTA chip, which was then flown through with the varying concentrations of analyte proteins without His-tags. The results showed that Mettl3 bound to Nsun4 and Ythdf2 (Fig. 5a, b). Additionally, Ythdf2 interacted with Nsun4 (Fig. 5c). The direct binding forms of the three members revealed that Mettl3, Nsun4 and Ythdf2 form a complex. Meanwhile, eEF1α-1 was shown to interact with Ythdf2 (Fig. 5d). The interaction model of the Nsun4, Mettl3, Ythdf2 and eEF1α-1 is shown in Fig. 5e. These data revealed model of the assembly of the NMYE complex in vitro.

**Nsun4 and Mettl3 promote chondrogenic differentiation progression in vivo**. A drilling-induced cartilage defect model in rats was used to study whether Nsun4 and Mettl3 promote chondrogenic differentiation in vivo (Fig. 6a)[5]. The surface of the

defects could be observed over time, and the results showed that the matrix and empty virus groups could not form fibrous tissue to repair the cartilage defect. Conversely, the OE-Nsun4 and OE-Mettl3 groups formed cartilage-like tissue (Fig. 6b).

Immunofluorescence detection assays showed that the Mettl3 and Nsun4 were successfully overexpressed in rat bone tissue (Fig. 6c, d). Compared to that in the empty virus- and matrix-treated groups, the expression of Sox9 was higher in the OE-Nsun4 and OE-Mettl3 treated groups (Fig. 6e). Meanwhile, the expression of the other cartilage marker genes, including Aggrecan (Supplementary Fig. 8a) and Col2 (Supplementary Fig. 8b), were also higher in the OE-Nsun4 and OE-Mettl3 groups. These data demonstrated that Nsun4 and Mettl3 promote chondrogenic differentiation in vivo Fig. 7.

**Discussion**

This study demonstrated that, after chondrogenic differentiation, m5C and m6A modification coregulate the translational reprogramming of Sox9. Nsun4 and Mettl3 co-methylate Sox9 by targeting on the 3'UTR region. Nsun4 and Mettl3 form a complex to regulate the translation of Sox9 by recruiting Ythdf2 and eEF1α-1.

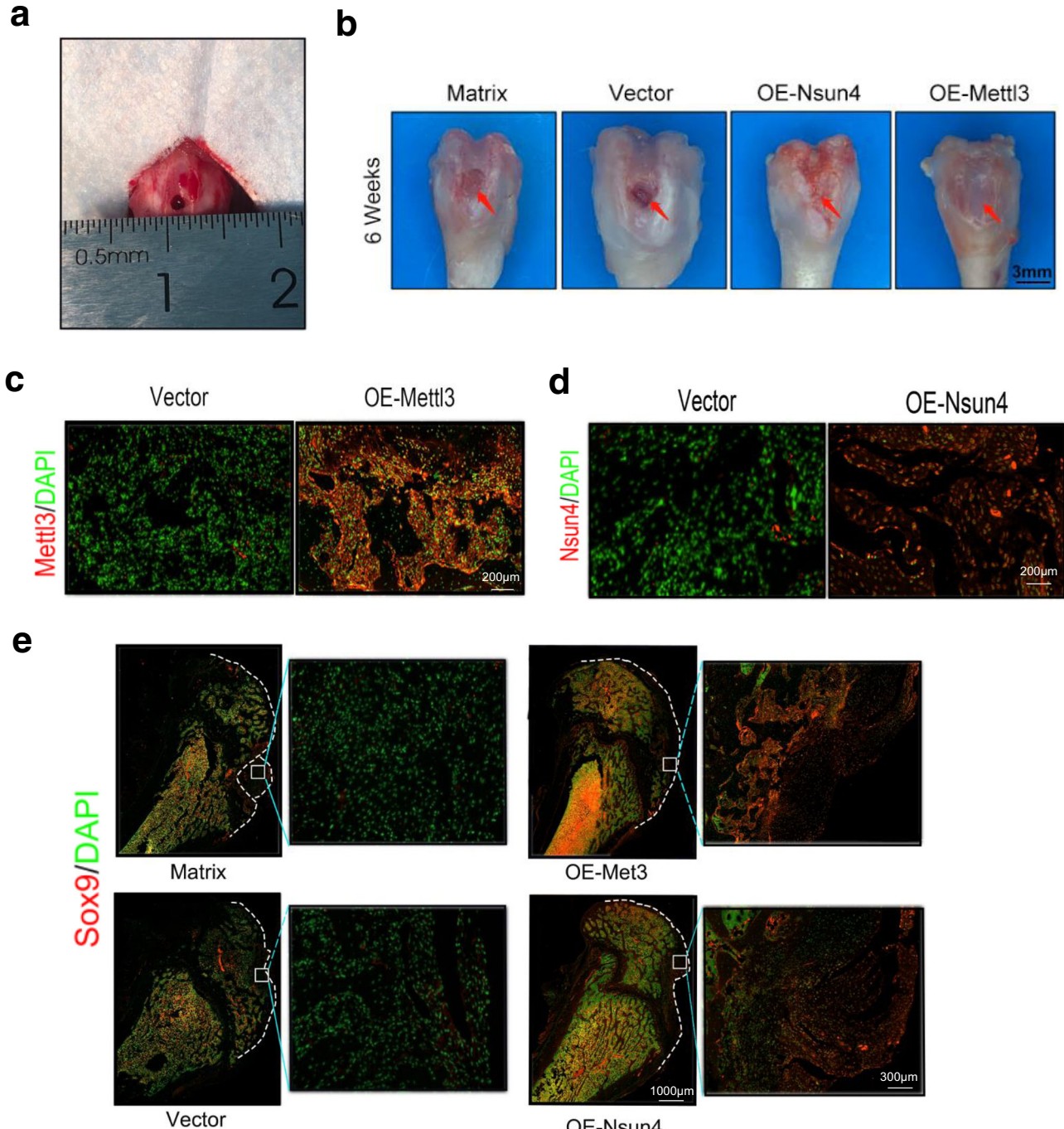

**Fig. 6 Nsun4 and Mettl3 promote chondrogenic differentiation progression in vivo. a** A 3-mm chondral defect was created in the medical area of the patella groove. **b** BMSCs infected vector/Nsun4-OE/Mettl3-OE adeno-associated virus for 7days, then were transferred to the defects for six weeks. Cartilage defects repaired in the Nsun4-OE and Mettl3-OE groups. Scale bars indicate 3 mm. **c**, **d** Verification of the efficiency of Nsun4 (**c**) and Mettl3 (**d**) overexpression at the protein level in BMSCs on 6 weeks after Nsun4-OE and Mettl3-OE adeno-associated virus infection, as detected by immunofluorescence. $n = 3$. **e** Sox9 protein expression of repaired cartilage visualized through immunofluorescence at six weeks.

The chondrogenic differentiation of BMSCs depends on Sox9, the main transcription factor of this process[14]. It has been reported that mTORC1 selectively controls the RNA translation of *Sox9* in skeletogenesis[36]. SIRT1 is also responsible for promoting chondrogenesis by increasing Sox9 nuclear localization[37]. Our data revealed higher Sox9 protein expression in chondrogenic pellets. Translation, the process by which a ribosome reads an mRNA template to guide protein synthesis, is a critical step in gene expression. Ribosome profiling is a deep sequencing-based tool that enables the detailed measurement of translation globally. At the core of this approach is to measure the density of ribosome-protected fragments (ribosome footprints) on a given transcript that provides a proxy for the rate of protein synthesis. The distribution of ribosome footprints can provide insights into translational efficiency[38]. The pattern of the ribosome footprint coverage of genes was found to be increased, especially on *Sox9*, which indicated enhanced translation of *Sox9*. However, whether m5C and m6A modifications regulate *Sox9* has received relatively

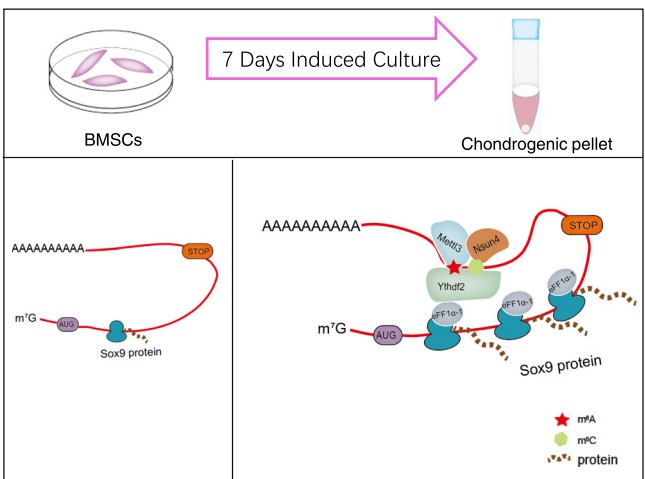

**Fig. 7 Schematic diagram showing the mechanism of the translation reprogramming of Sox9.** During chondrogenic differentiation of BMSCs, Nsun4 and Mettl3 co-methylated Sox9 by targeting on its 3'UTR region. Nsun4 and Mettl3 was assembled with recruitment of Ythdf2 and eEF1a-1. Eventually, the NMYE complex, carrying the methylated Sox9 mRNA, bound to ribosome, and then Sox9 mRNA was delivered to the ribosome for translation.

little attention. Our data revealed that the levels of $m^5C$ and $m^6A$ were increased remarkably in chondrogenic pellets, as well as in *Sox9* mRNA. We uncovered that $m^5C$ and $m^6A$ modifications are relevant to Sox9 translation after chondrogenic differentiation progression.

Recently, either $m^5C$ or $m^6A$ methylation has been increasingly reported to be involved in promoting stem cell differentiation[27]. Our study has demonstrated that $m^5C$ RNA methylation is regulated by Nsun4 and $m^6A$ depends on Mettl3 after chondrogenic differentiation of BMSCs. Deletion studies revealed that Nsun4 and Mettl3 are indispensable and play essential roles in regulating the progression of chondrogenic differentiation. Moreover, accumulated references have revealed that RNA modifications in 3'UTRs regulates mRNA looping to bring the 3' and 5' ends of the transcript in close proximity, which results in enhanced translation[39]. It has been reported that Mettl3 influences translation enhancement when it is tethered to the 3'UTR of a reporter mRNA[40]. Similarly, $m^5C$ in the 3'UTR of the cell cycle regulators CDK1 and p21 was shown to promote translation of these mRNAs[41,42]. Here, we first uncovered that Nsun4 and Mettl3 indispensably enhance Sox9 translation by forming a complex after chondrogenic differentiation. The complex targets Sox9 by binding to the 3'UTR. More importantly, the nucleotide 2062 of *Sox9* mRNA was confirmed to be the $m^5C$ methylation site through bisulfite sequencing. Subsequently, the nucleotide 2030 of *Sox9* was found to be the $m^6A$ methylation site via RIP-qPCR after chondrogenic differentiation. It has been reported that RNA modifications may act to stabilize or disrupt RNA secondary structure and influence RNA binding protein (RBP) accessibility, which may facilitate the promotion of RNA translation[43]. We found that the two sites (nucleotide 2030 for $m^5C$; nucleotide 2062 for $m^6A$) are close enough in the RNA structure that it is possible for the complex to co-methylate *Sox9* mRNA. However, whether the Nsun4/Mettl3 complex co-methylates at the two sites to regulate the RNA structure and translation needs to be further studied.

Ythdf2, characterized as an RNA methylation reader protein, modulates the life cycle of RNA. It has been reported that Ythdf2 acts as a regulator of mRNA translation and degradation by binding mRNAs with $m^6A$ methylation marks[44–46]. A recent study showed that Ythdf2, as an $m^5C$ reader protein could modulate rRNA maturation in human cells[20]. Our data revealed that Ythdf2 binds to the 3'UTR of *Sox9* mRNA after chondrogenic differentiation and regulates the protein expression of Sox9. Protein synthesis on the ribosome is promoted by the action of several translational GTPase factors. Multiple copies of a ribosomal protein, the so-called stalk protein, play a crucial role in the recruitment of translational GTPase factors to the ribosome[47]. eEF1α-1, an elongation factor, transports aminoacyl-tRNA to the ribosome in its GTP-bound form[48]. Our data show that Nsun4 and Mettl3 recruit eEF1α-1 in BMSCs undergoing chondrogenic differentiation. There are two possible mechanisms for the recruitment of eEF1α-1. One is that eEF1α-1 binds to ribosomes, and the Nsun4/Mettl3/Ythdf2 complex then recruits eEF1α-1 containing ribosomes. Another is that eEF1α-1 and Nsun4/Mettl3/Ythdf2 form a complex, and the ribosome interacts with the complex by binding with eEF1α-1. As a result, the NMYE complex, carrying the methylated *Sox9* mRNA, binds to ribosomes, and then *Sox9* mRNA is delivered to the ribosome for translation. Furthermore, our results showed that NMYE complex had an important effect on Sox9 protein expression, but not on mRNA expression. Overall, these findings indicate that the NMYE complex co-regulates the translational reprogramming of *Sox9*.

Interestingly, western blot assays showed that the total protein level of Nsun4 was downregulated, while the $m^5C$ level of the total mRNA, particularly *Sox9*, was increased. Using co-IP assays, we demonstrated that Mettl3 and Nsun4 formed a complex, and increased levels of Mettl3 recruited more Nsun4 resulting in protection from degradation after chondrogenic differentiation, which may play an effective regulatory role in RNA methylation.

There is limitation in this study. Enhancement of Nsun4/Mettl3 levels in stem cells promoted the repair of cartilage defects, however, a long-term observation in animals requires further studies.

In summary, we provide compelling in vitro and in vivo evidence demonstrating that $m^5C$ and $m^6A$ can co-regulate the progression of chondrogenic differentiation of BMSCs and *Sox9* translation reprogramming. The mechanism is shown in Fig. 7. Our present research showed that the mechanism of co-regulation of chondrogenic differentiation of BMSCs is through multiple RNA modifications working together. Nsun4 and Mettl3 could be targets for drug research to promote the chondrogenic differentiation of BMSCs for their use in the treatment of human cartilage defect. The enhancement of Nsun4/Mettl3 expression might be a strategy for replacement therapy for the treatment of human cartilage defects.

## Methods

**Cell culture**. Under sterile conditions, primary BMSCs were isolated and collected by flushing the femurs and tibias with alpha minimal essential medium (Gibco). Isolated BMSCs were expanded in SD rat mesenchymal stem cell medium (Syagen). The medium was changed once every 2–3 days and the third passage (P3) BMSCs were used at P3 for the following experiments.

**Chondrogenic differentiation of BMSCs**. P3 BMSCs were cultured in a monolayer to 95% confluence. After trypsinization, the cells were suspended in Dulbecco's Modified Eagle Medium (DMEM) low-glucose medium, and pelleted by centrifugation at 1500 rpm for 5 min. The supernatant was aspirated and the cells were resuspended in fresh DMEM low-glucose medium with 10 ng/ml TGF-β3. For pellet culture, $2 \times 10^5$ cells were centrifuged at 1500 rpm for 5 min in a 15 ml conical tube (Corning), and then incubated at 37 °C for 7 days in humidified air with 5% (v/v) $CO_2$.

**Cell transfection**. siRNAs targeting eEF1α-1, Ythdf2, Mettl3, Nsun4 and Nsun6 were designed and synthesized by RiboBio (Guangzhou, China). The siRNAs sequences are listed in Supplementary Table 2. The BMSCs ($2 \times 10^5$ cells/ml) were transfected with 10 nM siRNA using Lipofectamine 3000 (Invitrogen, USA)

according to the manufacturer's instructions. Adeno-associated virus (AAV) overexpressing Nsun4 and Mettl3 was designed and synthesized by GenePharma (Shanghai, China). The final volume of adenovirus to culture medium for transfection was 20 μL (the titre was $1 \times 10^{10}$ plaque-forming units [PFU]/mL). After incubation for 48 h in the medium, the medium was changed to induction for 5 days and subsequent experiments were performed.

**Western blot (WB) analysis**. Protein was extracted from the cells and chondroblasts by RIPA buffer (Thermo) with 100 × protease inhibitor cocktail (Thermo). The supernatant was separated by 10% sodium dodecyl sulphate polyacrylamide gel electrophoresis (SDS-PAGE). The target proteins were blotted onto polyvinylidene difluoride (PVDF) membranes (0.22 μm, Millipore, Germany). The membranes were blocked with 5% milk, and then incubated with primary rabbit anti-β-actin(Abcam, ab8226, 1:20000), rabbit anti-Sox9 (Abcam, ab185966, 1:1000,), rabbit anti-Aggrecan (Abcam, ab3788, 1:500), mouse anti-collagen II (Abcam, ab185430, 1:1000), rabbit anti-Mettl3 (Abcam, ab195352, 1:1000), rabbit anti-Nsun4 (Novus, NBP2-19594, 1:1000,), rabbit anti-eEF1a-1 (Santa, sc-21758, 1:2500), rabbit anti-YTHDF1 (Abcam, ab157542, 1:1000), rabbit anti-YTHDF2 (Millipore, ABE542, 1:1000,), and rabbit anti-YTHDF3 (Abcam, ab83716, 1:1000,) antibodies overnight at 4 °C. After incubation goat with anti-rabbit antibody (Abcam, ab6721, 1:20000) or goat anti-mouse antibody (Abcam, ab6789, 1:20000) conjugated with horseradish peroxidase (HRP), target proteins were visualized by enhanced chemiluminescence with a western blot detection kit (Millipore).

**Real-time quantitative PCR**. Total RNA extraction was performed with TRIzol Reagent (Life, 139505) and cDNA was synthesized using a cDNA reverse transcription kit (Roache). Amplification and detection were performed on a Light Cycler 480 Sybr System (Roche Applied Science) with the SYBR Green qPCR SuperMix (Promega) For quantitative RT-PCR, β-actin was used as an endogenous control. The primers are shown in Supplementary Table 1.

**RNA-seq**. Total RNA was extracted using a TRIzol reagent kit (Invitrogen, Carlsbad, CA, USA) according to the manufacturer's protocol. RNA quality was assessed on an Agilent 2100 Bioanalyzer (Agilent Technologies, Palo Alto, CA, USA) and examined by RNase free agarose gel electrophoresis. After total RNA was extracted, mRNA was enriched by oligo(dT) beads using the Ribo-Zero™ Magnetic Kit (Epicentre, Madison, WI, USA). Then, the enriched mRNA was fragmented and reverse transcribed into cDNA with random primers. Next, the cDNA fragments were purified, end repaired, "A" bases were added, and they were ligated to Illumina sequencing adapters. The ligation products were selected by agarose gel electrophoresis, PCR amplified, and sequenced using an Illumina Novase6000 by Gene Denovo Biotechnology Co. (Guangzhou, China).

**Ribo-seq**. The ribosomal profiling technique was carried out as reported previously[49], with a few modifications as described below. To prepare RFs, RNase I and DNA I were added to the lysate and incubate for 45 min at room temperature. Size exclusion columns were equilibrated with polysome buffer by gravity flow. Digested RFs were added to the column and centrifuged. Next, 10 μL 10% SDS was added to the elution, and RFs with a size greater than 17 nt were isolated according to the RNA Clean and Concentrator-25 kit (Zymo Research; R1017). rRNA was removed using a method reported previously[50]. Briefly, short (50–80 bases) anti-sense DNA probes complementary to the rRNA sequences were added to a solution containing RFs, and RNase H and DNase I were added to digest the rRNA and residual DNA probes. Finally, RFs were further purified using magnetic beads. After obtaining the ribosome footprints above, Ribo-seq libraries were constructed using the NEBNext[R] Multiple Small RNA Library Prep Set for Illumina. Briefly, adapters were added to both ends of the RFs, followed by reverse transcription and PCR amplification. The 140–160 bp size PCR products were enriched to enriched to generate to a cDNA library and sequenced using Illumina HiSeq X10 by Gene Denovo Biotechnology Co. (Guangzhou, China).

**Dot-blot assay**. mRNA was purified using a Dynabeads™ mRNA Purification Kit (Invitrogen). After 5 min of denaturation at 65–80 °C, equal amounts of serially diluted mRNA were added to a Hybond-N + membrane and crosslinked in a Stratalinker 2400 UV crosslinker twice using the Autocrosslink mode. The membrane was blocked with 5% BSA in 1 × PBST for 1 h at room temperature and incubated with anti- m6A (SySy, 202 003, 1:1000) and anti-m5C (Abcam, ab186830, 1:1000) antibody dilution buffer overnight at 4 °C with gentle shaking. After washing the membrane three times, the membrane was incubated with the horseradish peroxidase (HRP)-conjugated goat anti-mouse (Abcam, ab6789, 1:20000) or anti-rabbit antibody (Abcam, ab6721, 1:20000) for 1 h at room temperature. Membranes were developed with ECL Western Blotting Substrate to expose the signal.

**RIP-qPCR**. A Millpore RIP kit (Millipore) was used to examine m6A/m5C modifications and Nsun4/Mettl3/Ythdf2 interaction with genes according to the manufacturer's instructions. Cells were harvested and lysed in 110 μl of RIP lysis

buffer. Related antibody (5 μg) was added and incubated with magnetic beads, followed by rotation at RT for 30 min. The beads were washed with RIP wash buffer and resuspended in 900 μl of RIP buffer mixed with 100 μl of cell lysate. After rotation at 4 °C overnight, the beads were washed with wash buffer, followed by the digestion of proteins with proteinase K. RNA was purified by phenol chloroform extraction. The enriched RNA was fragmented into short fragments or not and analysed by RT- qPCR. The primers are shown in Supplementary Table 1.

**Protein co-immunoprecipitation and LC-MS**. The Pierce™ Direct Magnetic IP/Co-IP kit (Thermo Fisher; 88828) was used to examine the protein-protein interactions. Cells grown to 80–90% confluency in 10-cm dishes were lysed with lysis buffer. Proteins were immunoprecipitated from cell lysates with Nsun4 and Mettl3 antibodies and the corresponding IgG. After applying a magnet, proteins associated with protein A/G magnetic beads were washed three times and analysed by western blotting.

For mass spectrometry analysis, the IP proteins were resolved by SDS-PAGE, followed by sliver staining (Thermo Fisher). Each lane containing target proteins was digested with trypsin. A thermo Fisher Orbitrap Elite with Waters NanoAcuity UPLC was used to analyse the extracted peptides. MS data were searched against various sources (including GenBank, RefSeq, SwissProt, PDB, etc.) and the Mascot 2.3.02 software (Proteome Software) was applied for the data analysis. Peptides were filtered to a 1% FDR threshold.

**Bisulfite conversion of RNA and Sanger sequencing**. Bisulfite treatment was performed with EZ RNA methylation Kit (Zymo Research) according to the manufacturer's protocol. The target region of Sox9 was amplified by PCR using normal primers for untreated mRNAs and specific primers for bisulfite-treated mRNAs. To facilitate sequencing, the purified PCR products were ligated to the T vector using the pGEM-T easy vector systems kit (Promega). The ligation reaction was carefully transferred to the JM109 High Efficiency Competent Cells (Biomed), and cultured onto LB/ampicillin/IPTG/X-Gal plates. White colonies (at least 20 replicates for each sample) were selected for Sanger-based sequencing. The primers for the candidate fragment are shown in Supplementary Table 1.

**DNA constructs and luciferase reporter analysis**. The Sox9 mRNA 3'UTR ranging from nucleotides 1918 to 2320 was synthesized and cloned into the pmirGLO vector (Promega). A luciferase assay was performed using a dual luciferase reporter analysis system (Catalogue #E3971, Promega, USA) according to the manufacturer's instructions. Briefly, 293 T cells (Takara, 632180) were transfected with NC, siMettl3, siNsun4, and siYTHDF2 in a 24-well plate. After transfection for 12 h, each cell line was transfected with pmirGLO, or pmirGLO-Sox9 3'UTR. After a 48 h incubation, cells were subjected to analysis with the Dual-Glo Luciferase System (Promega).

**Protein expression and purification**. The cDNAs encoding Nsun4, Mettl3, eEF1a-1 and Ythdf2 were cloned into the NdeI and XhoI sites of the pET28a vector. His-tagged Nsun4, Mettl3, eEF1α-1 and Ythdf2 were expressed in the E.coli strain BL21DE3 (TransGen Biotech; CD801-03). The bacteria were grown in L-Broth at 37 °C to an h ≈ 0.8, and then induced with 1 mM isopropyl-L-thio-B-D-galactopyranoside (IPTG) at 16 °C overnight. Bacteria were harvested and lysed in binding buffer (25 mM Tris, 500 mM NaCl, pH 8.0). Proteins were purified on an NTA nickel-agarose column (NUPTEC, NRPB07L-500) and eluted with an imidazole gradient. The purified proteins were then cleaved with sumo protease at 25 °C. Then another round of NTA-nickel-agarose chromatography was performed to remove the cleaved residues containing the His tag. The final preparations of purified proteins were dialyzed against HBS-P+ running buffer (Cytiva, BR100669).

**SPR experiments**. All SPR experiments were carried out at 25 °C on a Biacore 8 K instrument (GE Healthcare), using mono NTA-coated biosensor chips (GE Healthcare) and HBS-P+ as running buffer. Flow cell2 (FC2) was used to capture the ligand protein. Flow cell1 (FC1) was the reference channel. The NTA chips were coated with 0.5 mM NiCl2, and then His-tagged proteins were captured on the chips. The protein capture level was set at 100–150 RU. Analyte protein solutions ranging from 3–100 nM were injected for 100 sec, and then dissociated in running buffer for 150 s. At the end of the dissociation period the sensor chip was regenerated to remove any remaining bound material by injecting 350 mM EDTA for 120 s.

**Animal surgical procedure**. Skeletally mature Sprague-Dawley (SD) rats averaging 190–210 g in weight were purchased from Guangdong Medical University. All animal experimental procedures were approved by the Experimental Ethics Committee of Guangdong Medical University. The rats were anaesthetized via strictly aseptic techniques during surgery. The knee joint was opened laterally by dislocating the patellar tendon, and a full thickness osteochondral defect (3-mm in diameter) was created in the trochlear groove of the femur. Then, approximately $10^7$ BMSCs were mixed with 1 ml hydrogel matrix (Coring, Cat#354250) and were injected into the defect. Penicillin G (80,000 units/rat, diluted on 0.9% sodium

chloride, i.p) was administered to avoid infection postoperatively. The animals were sacrificed by intravenous injection of a solution 6 weeks after the operation, and the repaired cartilage was taken for evaluation.

**Immunofluorescence staining**. The repaired cartilage was embedded in paraffin and cut into 5 μm sections. The sections were dewaxed and permeabilized in 0.25% Triton X-100 in PBS for 30 min at RT. Next, the sections were blocked with 5% BSA in PBS for an hour at RT and were incubated overnight with each indicated primary antibody at 4 °C. The next day, the cells were washed with PBS three times, and then Alex-647-conjugated Goat anti-rabbit or mouse secondary antibodies were incubated for 1 h at RT. Nuclei were stained with diamidino-2-phenyl-indole-dihydrochloride (DAPI; Solabo). The primary antibody dilutions were 1:500 for Sox9 (Abcam), 1:100 for Aggrecan (Abcam), and 1:250 for Col2 (Abcam) and secondary antibody dilution was 1:1000. Fluorescence images were obtained by Gen5 3.05.

**Statistics and reproducibility**. Data are given as means±standard error of the mean. Statistical analysis was performed using Student's $t$ test for two group comparisons and one-way analysis of variance (ANOVA) for multiple group comparisons (GraphPad Software, San Diego. CA). Numbers of replicates are described in detail in the figure legends. The data were considered statistically significant when $*p < 0.05$, $**p < 0.01$, and $***p < 0.001$, ns = not significant.

**Reporting summary**. Further information on research design is available in the Nature Research Reporting Summary linked to this article.

## Data availability

The data generated or analyzed during the current study are included in this published article (and its Supplementary Information files). Source data underlying the graphs can be found in Supplementary Data 1. RNA sequencing data and Ribosome sequencing data were deposited in the Sequence Read Archive (SRA) with the accession number SUB11204281 (RNA sequencing) and SUB11251451(Ribosome sequencing). The mass spectrometry-based proteomics data were obtained from iProx database with the number IPX0004238001. The plasmid have been deposited to Addgene with the ID 183091. All other data that support the finding of this study are available from the corresponding author on reasonable request.

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

## Acknowledgements

This research was supported by the Natural Science Foundation of Guangdong Province (2017A030312009), China Postdoctoral Science Foundation (2021M700961), Sanming Project of Medicine in Shenzhen (SZZYSM 202106006), Bao'an TCM Development Foundation (2020JD441, 20190509084742569), National Administration of Traditional Chinese Medicine (GZY-FJS-2019-001, GZY-FJS-2019-002). We thank Natural Science Foundation of Guangdong Province and Science, Technology Innovation Foundation of Shenzhen and Scientific Research Fund of the State Administration of Traditional Chinese Medicine for supporting our research.

## Author contributions

L.Y., Z.R., S.Y. performed the experiments and data analysis. J.L., Z.L., S.Y. aided culture cells and cell associated experiments. L.Z., A.L., X.L. provided technical help. Z.R., L.Z., J.G., W.Z., W.K., H.L., and D.C. designed and directed the research. L.Y. wrote the manuscript.

## Competing interests

The authors declare no competing interests.
