## [Peer Review File · Communications Biology]

Reviewers' comments:

Reviewer #1 (Remarks to the Author):

The manuscript by Lin Yang and co-authors elucidates the molecular mechanism involving the Sox9 transcription factor to mediate chondrogenic differentiation. In particular, the authors describe an increase of the 5-methylcytosine (m5C) and N6-methyladenosine (m6A) levels in the 3'-UTR region of Sox9 mRNA and suggest that Nsun4-mediated m5C and Mett13-mediated m6A are required for Sox9 -regulated chondrogenic differentiation.

The manuscript is interesting and the study correlating the methylation of Sox9 mRNA and the chondrogenic differentiation of bone marrow mesenchymal stem cells is new. However, in the current form, the manuscript presents several flaws that must be addressed.

Major points

1. The results section must be significantly revised.

The authors performed many experiments also with innovative technology. However, all results are poorly described and difficult to follow.

For instance, experiments showing the binding of Nsun4 and Mett13 to the 3' UTR region of Sox9 mRNA are superficially documented.

Similarly, the results showing the m5C and m6A co-methylation are almost listed.

Moreover, each paragraph lacks a small introduction describing the rationale of the presented experiments.

A general schematic of the rationale of the work, might useful.

2. Experiment reporting the assembly model of NMYE complex proteins must be validated also in the chondrogenic pellet (as the authors claim in the discussion)

3. In vivo experiments are still preliminary. Experiments must be performed for a longer time and foremost in a more appropriate animal model.

Minor points

1- The abstract must be better focused on the mechanism of methylation of Sox9

2- The introduction must be improved. For instance, the name of the RNA methyltransferases Nsun2, 3, 4, 5, 6, and 7 must be clearly indicated.

Reviewer #2 (Remarks to the Author):

In this paper, the authors provide evidence for a role of Nsun4 and Mett13 mediated cytosine and adenine RNA modification in chondrogenic differentiation. The findings are novel and potentially interesting, but I'm concerned about the conclusiveness of the RNA modification analysis (see detailed comments below). Additional experiments and controls need to be provided to firmly establish the mechanistic base of the paper. Also, I think that it is important to correct the scientifically inappropriate use of the term "epigenetic".

1. The dotblots (Fig. 1E-F, 2H, 3C-D) lack important controls. The purity of the mRNA preparations needs to be confirmed by proving the absence of detectable tRNA and rRNA. Furthermore, the induction of m5C and m6A needs to be confirmed by masspec analysis.

2. Experimental data needs to be shown to confirm the presence and establish the specificity of the A2030 methylation signal (Fig. S3A).

3. The bisulfite sequencing results (Fig. 3K-M) lack important controls and need to be shown in full detail. How many replicates were analyzed? Also, methylation ratios need to be provided for all cytosine residues of the PCR amplicon to establish the specificity of the C2062 methylation signal. It should also be noted that the bisulfite primers shown in Tab. 1 are not bisulfite primers. If these primers were indeed used for the analysis, the results would be certain to represent bisulfite

conversion artifacts.

4. The Sox9 3'-UTR reporter assay (Fig. 4G) provides an excellent opportunity to confirm the functional relevance of the A2030 and C2062 modifications. For this, the authors need to generate and test mutated constructs that cannot be methylated (e.g. A2030G and C2062T).

5. The authors incorrectly use the term "epigenetic" to describe RNA modifications and their effects. This needs to be changed in the title and throughout the text.

6. English language editing is required throughout the manuscript.

Reviewer #3 (Remarks to the Author):

In this manuscript entitled "The complex of Nsun4 and Mettl3 epigenetically co-regulate the translation reprogramming of Sox9 to promote chondrogenic differentiation of BMSCs" Yang et al found enhanced translation as well as increased m5C and m6A level of mRNAs including essential transcription factor Sox9 during chondrogenic differentiation. Mechanistically, they found that Nsun4 and Mettl3 can interact with each other and recruit Ythdf2 and eEF1a-1 to form a complex at 3'UTR of Sox9 to regulate translation of Sox9. At last they show overexpression of Nsun4 or Mettl3 can promote chondrogenic differentiation in vivo using drilling-induced cartilage defect model suggesting a potential clinical relevance.. Overall, this study characterized the role epi-transcriptional modifications in regulating chondrogenic differentiation of BMSC and identified a novel pathway that Nsun4-m5C and Mettl3-m6A co-regulate translation.

However, the results presented by the authors regarding Nsun4-m5C's contribution to chondrogenic differentiation seems to be contradictory to their proposed model. It has already been reported that m6A is essential for BMSC's fate and this study expands the known results.

Several key questions need to be addressed

- 1) The authors claim that Nsun4 is accounting for m5C regulation in chondrogenic differentiation. They showed decreased Nsun4 expression after induction (Fig 2A-B, S1D) and knockdown of Nsun4 reduced m5C level (Fig 2H and S1H). However, the m5C level is increased during chondrogenic differentiation (Fig 1E). Can authors explain these opposite results?
- 2) In Fig 1B and C, the authors show increased translation and protein abundance of some important genes during chondrogenic differentiation. What about RNA level of these genes? Does RNA level of these genes change? This is important to understand whether the change of protein is only due to translation or not.
- 3) In Fig S1G, the knockdown of Mettl3 is not efficient. It's not clear how such a modest decrease of Mettl3 results in a dramatic phenotype.
- 4) What's the expression of other m6A regulators especially writers including Mettl14, Wtap, Rbm15 and Zc3h13 during chondrogenic differentiation?
- 5) It is surprising that Nsun4 knockdown reduced m6A levels and Mettl3 knockdown reduced m5C level. What is the expression of Mettl3 upon Nsun4 knockdown and the expression of Nsun4 upon Mettl3 depletion? Besides, it's not consistent with results showing decreased Nsun4 expression and increased m6A level during chondrogenic differentiation. The authors should O/E Mettl3 or NSUN4 and then KD these proteins to determine if they are dependent. What happens to the proteins NSUN4 and METTL3 after KD of each and do they then stabilize each other?
- 6) What happens to NSUN4 and METTL3 abundance after YTHDF2 depletion?
- 7) In Figure 3, Mettl3 and Nsun4 can interact with each other only under induction condition. Why are they not interacting in the basal state? Same question is also observed in Figure 4 with Ythdf2 and eEF1a. Do these interactions depend on RNA since authors proposed a model that they assemble a complex on 3'UTR of Sox9?
- 8) In Fig 3J, there are stronger peaks on Sox9 exon than 3'UTR. The coding region should also be tested.
- 9) In Fig 3K-M, RIP experiments are used to determine that the m6A sites are direct but without crosslinking or mutating the sites its not clear if these mapped sites are correct.

- 10) Fig 3M is confusing, it remains unclear how the sequence frequency logo translates into reduction of m5C methylation upon knockdowns without statistical analysis.
- 11) What happens if you mutant potential m6A or m5C sites on the 3'UTR in the reporter assay?
- 12) Since Nsun4 is reduced and Sox 9 is increased during chondrogenic differentiation, it's surprising that overexpression of Nsun4 increase Sox9 expression. How can they explain these results? What's the m5C and m6A level upon Nsun4 or Mettl3 overexpression.
- 13) The authors should explain more for the reader what it means if there is more distribution of ribosomes on the transcripts and how does that relate to an increase in translation. What about translational efficiency relative to global mRNA levels?
- 14) The overall logic and rationale for the study should be improved. Why are these modifications chosen and why SOX9 focused on and not the other targets?
- 15) In Figure 2 the images of KD of METTL3 and NSUN4 seem different to the controls but are different between each other. A more careful assessment of what stage of differentiation is blocked or what are these cells?
- 16) The authors should show in vivo that KD of these factors can block chondrocyte differentiation or effect regeneration not just improve it with overexpression.
- 17) Some form of pathway analysis could be used to strengthen their conclusions about cell programs and cell types.

Reviewer #1 (Remarks to the Author):

The manuscript by Lin Yang and co-authors elucidates the molecular mechanism involving the Sox9 transcription factor to mediate chondrogenic differentiation. In particular, the authors describe an increase of the 5-methylcytosine (m5C) and N6-methyladenosine (m6A) levels in the 3'-UTR region of Sox9 mRNA and suggest that Nsun4-mediated m5C and Mettl3-mediated m6A are required for Sox9 -regulated chondrogenic differentiation. The manuscript is interesting and the study correlating the methylation of Sox9 mRNA and the chondrogenic differentiation of bone marrow mesenchymal stem cells is new.

However, in the current form, the manuscript presents several flaws that must be addressed.

Major points

1. The results section must be significantly revised.

The authors performed many experiments also with innovative technology. However, all results are poorly described and difficult to follow.

For instance, experiments showing the binding of Nsun4 and Mettl3 to the 3'UTR region of Sox9 mRNA are superficially documented. Similarly, the results showing the m5C and m6A co-methylation are almost listed. Moreover, each paragraph lacks a small introduction describing the rationale of the presented experiments.

A general schematic of the rationale of the work, might useful.

Response:

Thanks for your positive comments. This section has been revised. The results showing the binding of Nsun4 and Mettl3 to the 3'UTR region of Sox9 mRNA and the co-methylation of m⁵C and m⁶A have been modified. The small introduction of each paragraph has been supplemented. Please kindly see Line115-116 on page 4, Line 123-124 on page 4, Line133-135 on page 4, Line186-192 on page 6, Line 213-214 on page 6.

2. Experiment reporting the assembly model of NMYE complex proteins must be validated also in the chondrogenic pellet (as the authors claim in the discussion)

Response:

We are awfully sorry for the carelessness of writing that and the assembly model of NMYE complex has been validated in the chondrogenic pellet already. Please kindly see Figure 3A-B, Figure 4A-D. Also, the limitation in the discussion has been deleted.

3. In vivo experiments are still preliminary. Experiments must be performed for a longer time and foremost in a more appropriate animal model.

Response:

Thanks for the comments. Cause of the limited time and conditions, we have planned a long-term observation on animals for further studied.

Minor points

1- The abstract must be better focused on the mechanism of methylation of Sox9

Response:

The abstract has been revised. The elucidation focused on the methylation mechanism of Sox9 has been emphasized. Please kindly see Line34-42 on page1-2.

2- The introduction must be improved. For instance, the name of the RNA methyltransferases Nsun2, 3, 4, 5, 6, and 7 must be clearly indicated.

Response:

Thanks for the comments. The introduction has been revised. The name of the RNA modification regulators has been clearly indicated. Please kindly see Line 83-94 on page3. Moreover, the language editing in this manuscript has been improved by native speakers. The certificate of editing is attached as Supplementary material 1.

Reviewer #2 (Remarks to the Author):

In this paper, the authors provide evidence for a role of Nsun4 and Mettl3 mediated cytosine and adenine RNA modification in chondrogenic differentiation. The findings are novel and potentially interesting, but I'm concerned about the conclusiveness of the RNA modification analysis (see detailed comments below). Additional experiments and controls need to be provided to firmly establish the mechanistic base of the paper. Also, I think that it is important to correct the scientifically inappropriate use of the term "epigenetic".

1. The dotblots (Fig. 1E-F, 2H, 3C-D) lack important controls. The purity of the mRNA preparations needs to be confirmed by proving the absence of detectable tRNA and rRNA. Furthermore, the induction of m5C and m6A needs to be confirmed by masspec analysis.

Response:

① We have supplemented the methylene blue image as controls for the dot blots. Please kindly see: Figure 1E-F, Figure 2H-I, Figure 3C-D, Supplementary Figure 2G-H, Supplementary Figure 3G-J.

②

To confirm the purity of the mRNA, the total RNA and purified RNA samples run on an Agilent 2100 Bioanalyzer. The left fluorescence plot was the electropherogram of the total RNA with 18S and 28S ribosomal peaks, and the peaks were disappeared in the purified RNA as shown in the right plot.

3)

The data of mass spectrometry has been supplemented. The results showed that the m⁵C and m⁶A levels were significantly increased in chondrogenic pellet. Please kindly see: Line 126-127 on page 4 and Supplementary Figure 1.

2. Experimental data needs to be shown to confirm the presence and establish the specificity of the A2030 methylation signal (Fig. S3A).

Response:

To confirm the specificity of the A2030 methylation signal, we cloned the 300-nucleotide (nt)-long WT or mutant 3'UTR truncation sequence (from 1950 to 2250) into the pmirGLO luciferase (Luc) reporter (A). Knockdown of Mettl3 decreased the Luc activity and overexpressed Mettl3 increased the Luc activity

but not that of the mutant reporter, which further confirmed that nucleotide 2030 was m6A methylation site (B).

The data has been supplemented. Please kindly see: Line 202-207 on page 6 and Figure S4C-D.

3. The bisulfite sequencing results (Fig. 3K-M) lack important controls and need to be shown in full detail. How many replicates were analyzed? Also, methylation ratios need to be provided for all cytosine residues of the PCR amplicon to establish the specificity of the C2062 methylation signal. It should also be noted that the bisulfite primers shown in Tab. 1 are not bisulfite primers. If these primers were indeed used for the analysis, the results would be certain to represent bisulfite conversion artifacts.

Response:

1) 20 replicates were analyzed for each group, and data are presented from three independent experiment. Please kindly see: Line 475-477 on page 13.

2)

Untreated Results	Treated Results
CAGCTCACCAGACCCTGAGGAGAC	TAGTTTATTAGATTTTGAGGAGAT
CTTGAAGAGCAATGGTGACAGAGTT	TTTGAAGAGTAATGGTGATAGAGTT
GATCTTTTTTTTTTTTTTTTTTAAGAA	GATTTTTTTTTTTTTTTTTTAAGAA
GAAAAGGAAAAAGAAAACGCTGAA	GAAAAGGAAAAAGAAAATGTTGAA
GAAAATCAAGAACCAATTGAAATTC	GAAAATTAAGAATTAATTGAAATTT
CTTTGGACACTTTTTTTTTTTGGTCTG	TTTTGGATATTTTTTTTTTTGGTTTG
TCGTTATTTTTAAAGATGTAAGTGA	TTGTTATTTTTAAAGATGTAAGTGA
AGGTAACGATTGCTGAGATTCCAGG	AGGTAATGATTATTGAGATTTTAGG
AGAGAGACTTTAAGACATTATCTGA	AGAGAGATTTAAGATATTATTGA
GCTCATGCAAACACGTTGCAAATGG	GTTTATGTAAATACGTTGTAAATGG
CCCGGCCATTCGTGGCCAGATGGAC	TTTGGTTATTTGTGGTTAGATGGAT

The PCR amplicon of the region of interest was shown, and the methylation ratios was 98%. The result showed the specificity of the C2062 methylation signal.

3) We are awfully sorry for the carelessness. Now, we have revised the bisulfite primers. Please kindly see: New Table 1.

4. The Sox9 3'-UTR reporter assay (Fig. 4G) provides an excellent opportunity to confirm the functional relevance of the A2030 and C2062 modifications. For this, the authors need to generate and test mutated constructs that cannot be methylated (e.g. A2030G and C2062T).

Response:

To further confirm the functional relevance of the A2030 and C2062 modifications, we constructed the pmirGLO-Sox9 luciferase reporter by ligating Sox9 3'UTR wild type (WT) and mutant type (Mut) to the multiple cloning site (MCs) (A). The dual-luciferase assay showed that translation efficiency of Sox9 in siNsun4, siMettl3 and siYthdf2 groups were significantly downregulated than that in 293T NC group in WT reporter, and was not changed in mutations of the m⁵C and m⁶A sites (B).

The data has been supplemented. Please kindly see: Line 240-245 on page 7 and Figure 4H-I.

5. The authors incorrectly use the term "epigenetic" to describe RNA modifications and their effects. This needs to be changed in the title and throughout the text.

Response:

The title has been changed to "The Nsun4 and Mettl3 complex promotes chondrogenic differentiation of BMSCs by modulating the translation reprogramming of Sox9" and the term "epigenetic" has been deleted and revised throughout the text.

6. English language editing is required throughout the manuscript.

Response:

Thanks for the comments. The language editing in this manuscript has been improved by native speakers. The certificate of editing is attached as Supplementary material 1.

Reviewer #3 (Remarks to the Author):

In this manuscript entitled "The complex of Nsun4 and Mettl3 epigenetically co-regulate the translation reprogramming of Sox9 to promote chondrogenic differentiation of BMSCs" Yang et al found enhanced translation as well as increased m⁵C and m⁶A level of mRNAs including essential transcription factor Sox9 during chondrogenic differentiation. Mechanistically, they found that Nsun4 and Mettl3 can interact with each other and recruit Ythdf2 and eEF1a-1 to form a complex at 3'UTR of Sox9 to regulate translation of Sox9. At last they

show overexpression of Nsun4 or Mettl3 can promote chondrogenic differentiation in vivo using drilling-induced cartilage defect model suggesting a potential clinical relevance. Overall, this study characterized the role epitranscriptional modifications in regulating chondrogenic differentiation of BMSC and identified a novel pathway that Nsun4-m⁵C and Mettl3-m⁶A co-regulate translation.

However, the results presented by the authors regarding Nsun4-m⁵C's contribution to chondrogenic differentiation seems to be contradictory to their proposed model. It has already been reported that m⁶A is essential for BMSC's fate and this study expands the known results.

Several key questions need to be addressed

1) The authors claim that Nsun4 is accounting for m⁵C regulation in chondrogenic differentiation. They showed decreased Nsun4 expression after induction (Fig 2A-B, S1D) and knockdown of Nsun4 reduced m⁵C level (Fig 2H and S1H). However, the m⁵C level is increased during chondrogenic differentiation (Fig 1E). Can authors explain these opposite results?

Response:

Thank you very much for your professional question.

① On one hand, during chondrogenic differentiation, the total protein of Nsun4 was found decreased, however the very Nsun4 interacted with Mettl3 was increased significantly (Fig. 3A-B), thus the level of m⁵C appears to be increased. On the other hand, with silencing of Nsun4, the very Nsun4 interacted with Mettl3 decreased significantly, which reduced the m⁵C level. Taken together, the level of m⁵C was directly regulated by Nsun4 and Mettl3 complex.

②

It is known that m⁵Cs in RNAs are introduced by members of the NOL1/NOP2/SUN domain (Nsun) family, as well as the DNA methyltransferase (DNMT) homologue DNMT2. We supplemented RT-qPCR to detect the expression of *DNMT2*. The result showed that gene expression of *DNMT2* has not changed during chondrogenic differentiation, which further indicate that the methylation of m⁵C is dependent on Nsun4 during differentiation.

2) In Fig 1B and C, the authors show increased translation and protein abundance of some important genes during chondrogenic differentiation. What about RNA level of these genes? Does RNA level of these genes change? This is important to understand whether the change of protein is only due to translation or not.

Response:

The RNA level of Aggrecan, Col2 and Sox9 were increased during chondrogenic differentiation. However, Sox9 underlies chondrocyte differentiation by transcriptionally activating markers of overtly differentiated chondrocytes, such as aggrecan and Col2. Therefore, our research focuses on Sox9.

In this study, Sox9 mRNA expression was increased, but the Ribo-seq data showed that the translation level of Sox9 was also increased (Figure 1D). The change of protein is not only due to translation.

The data revealed that the NMYE complex regulates the translation of Sox9 (Figure 4H-I) and has no effect on its transcription (Figure 4J).

3) In Fig S1G, the knockdown of Mettl3 is not efficient. It's not clear how such a modest decrease of Mettl3 results in a dramatic phenotype.

Response:

Sorry to make you confused about the deletion effect of Mettl3. For the original result, the fragment 1-3 were transfected into BMSCs for 24h followed by WB found the fragment 3 has deletion effect. And in the subsequent experiments, the BMSCs were transfected with fragment3 for 48h and then induced for 7 days. To accurately value the silence effect of fragment3, we transfected it for 48h and confirmed that its deletion efficiency reached over 80% (see figure above). Also, we have replaced the original picture with the above new one. Please kindly see: New Supplementary Figure 2J.

4) What's the expression of other m6A regulators especially writers including Mettl14, Wtap, Rbm15 and Zc3h13 during chondrogenic differentiation?

Response:

m⁶A is regulated by Mettl3, Mettl14 and Wtap, as well as Rbm15 and Zc3h13. We quantified the expression of above genes, resulting in that the RNA level of Mettl3 and Rbm15 increased significantly, while Mettl14, Wtap and Zc3h13 has no obvious change.

Mettl3 acts as the only catalyst with an internal S-adenosyl methionine (SAM)-binding domain for transferring methyl groups in SAM to adenine bases in RNAs, indicating it is the most important component of the m⁶A methyltransferase complex to focus on. And RNA-binding motif protein 15 (Rbm15) is able to target specific RNA motifs to regulate m6A modification. So, for the next study, we may deeply focus on the RNA recognition protein including Rbm5 to explore the specific m6A regulator of chondrogenic genes[DOI: [org/10.1186/s12943-020-01204-7](https://doi.org/10.1186/s12943-020-01204-7)].

5) It is surprising that Nsun4 knockdown reduced m6A levels and Mettl3 knockdown reduced m5C level. What is the expression of Mettl3 upon Nsun4 knockdown and the expression of Nsun4 upon Mettl3 depletion? Besides, it's not consistent with results showing decreased Nsun4 expression and increased m6A level during chondrogenic differentiation. The authors should O/E Mettl3 or NSUN4 and then KD these proteins to determine if they are dependent. What happens to the proteins NSUN4 and METTL3 after KD of each and do they then stabilize each other?

Response:

①

To determine whether Mettl3 and Nsun4 are dependent on each other, we confirmed that the expression levels of Mettl3 and Nsun4 were lower in siNsun4 and siMettl3 cells than the NC groups during chondrogenic differentiation (Fig.A and B). Meanwhile, we individually upregulated Mettl3 and Nsun4 expression in BMSCs by AAV infection. As shown in Figure C and D, the Mettl3 and Nsun4 level was markedly increased in Mettl3 and Nsun4-overexpressing (OE) BMSCs. Consistently, there were significantly increases in Mettl3 and Nsun4 protein expression by Nsun4 and Mettl3 overexpression after chondrogenic differentiation (Fig.E and F).

The data of OE-Mettl3 or Nsun4 and KD these proteins has been supplemented. Please kindly see: Line 169-177 on page 5 and Supplementary Figure 3A-F.

② The result of CO-IP assay (Fig.3A-B) showed that the level of the interaction between Nsun4 and Mettl3 was increased after chondrogenic differentiation. Nsun4 formed a complex with Mettl3 and was required for regulating m⁶A level after chondrogenic differentiation.

6) What happens to NSUN4 and METTL3 abundance after YTHDF2 depletion?

Response:

After Ythdf2 depletion, we observed a decreased Nsun4 and Mettl3 protein level in si-Ythdf2 cells.

The data has been supplemented. Please kindly see: Line 232-234 on page 7 and Figure 4E.

7) In Figure 3, Mettl3 and Nsun4 can interact with each other only under induction condition. Why are they not interacting in the basal state? Same question is also observed in Figure 4 with Ythdf2 and eEF1a. Do these interactions depend on RNA since authors proposed a model that they assemble a complex on 3'UTR of Sox9?

Response:

①

In fact, this is a technique problem, cause the western blot exposure result is not lineage, which means the unseen blot is not presenting no proteins. For example, after 2 times gradient dilution, the last group cannot be observed clearly, while after removing the first two groups that expressed much higher, the last group was able to be exposed clearly again. For the same reasons, the last group of strips (Fig.3A-B, Fig.4A-D) showing nearly no blot which doesn't mean no proteins exists, but very low level of proteins, thus hard to be visualized.

② We have no evidence that the interactions depend on RNA, however the results of SPR experiment showed that the proteins can still interact between each other without RNA.

8) In Fig 3J, there are stronger peaks on Sox9 exon than 3'UTR. The coding region should also be tested.

Response:

RIP-qPCR showed that the Nsun4 and Mettl3 enrichment of Sox9 5'UTR, CDs and 3'UTR were significantly upregulated after induction. However, the enrichments in 3'UTR were the highest among three regions, suggesting that

Nsun4 and Mettl3 methylation in 3'UTR region might be the most dynamic.

9) In Fig 3K-M, RIP experiments are used to determine that the m6A sites are direct but without crosslinking or mutating the sites it is not clear if these mapped sites are correct.

Response:

To confirm that the 2030nt was m⁶A site, we cloned the 300-nucleotide (nt)-long WT or mutant 3'UTR truncation sequence (from 1950 to 2250) into the pmirGLO luciferase (Luc) reporter (A). Knockdown of Mettl3 decreased the Luc activity and overexpressed Mettl3 increased the Luc activity but not that of the mutant reporter, which further confirmed that nucleotide 2030 was m⁶A methylation site (B).

The data has been supplemented. Please kindly see: Line 202-207 on page 6 and Figure S4C-D.

10) Fig 3M is confusing, it remains unclear how the sequence frequency logo translates into reduction of m5C methylation upon knockdowns without statistical analysis.

Response:

In the sequence frequency logo, "C" represents the m⁵C modified site unchanged after bisulfite treatment, and "T" represents the changed C site. The data image has been changed to statistical analysis chart. Please kindly see: New Figure 3M.

11) What happens if you mutant potential m6A or m5C sites on the 3'UTR in the reporter assay?

Response:

To further confirm the functional relevance of the A2030 and C2063 modifications, we constructed the pmirGLO-Sox9 luciferase reporter by ligating Sox9 3'UTR wild type (WT) and mutant type (Mut) to the multiple cloning site (MCs) (A). The dual-luciferase assay showed that translation efficiency of Sox9 in siNsun4, siMettl3 and siYthdf2 groups were significantly downregulated than that in 293T NC group in WT reporter, and was not changed in mutations of the m⁵C and m⁶A sites (B).

The data has been supplemented. Please kindly see: Line 240-245 on page 7 and Figure 4H-I.

12) Since Nsun4 is reduced and Sox9 is increased during chondrogenic differentiation, it's surprising that overexpression of Nsun4 increase Sox9 expression. How can they explain these results? What's the m⁵C and m⁶A level upon Nsun4 or Mettl3 overexpression.

Response:

①As the hypothesis (Nsun4 and Mettl3 may formed a complex and bound to the Sox9 mRNA to regulate the chondrogenic differentiation of BMSCs) we proposed and evidences we exhibited (Fig.3A-B), the key role to regulate chondrogenic differentiation is the combined state of Nsun4 and Mettl3. Although Nsun4 is reduced during chondrogenic differentiation, the formation of complex between Nsun4 and Mettl3 showed increased, which leads upregulated m⁵C and m⁶A, thus promotes chondrogenic differentiation. Moreover, m⁵C and m⁶A RIP-qPCR has been identified that the m⁵C and m⁶A of Sox9 mRNA were remarkably increased after induction (Fig.1G-H), and the result of western blot confirmed the increased expression of Sox9 (Fig.1B).

②

Through performing Dot blot, we assessed the m⁵C and m⁶A level upon Nsun4 or Mettl3 overexpression. The results showed that m⁵C and m⁶A level were remarkably increased in OE-Mettl3 or OE-Nsun4 cells after chondrogenic differentiation.

The data of OE-Nsun4/Mettl3 has been supplemented. Please kindly see: Line 179-181 on page 5 and Supplementary Figure S3G-J.

13) The authors should explain more for the reader what it means if there is more distribution of ribosomes on the transcripts and how does that relate to an increase in translation. What about translational efficiency relative to global mRNA levels?

Response:

Translation, the process by which a ribosome reads an mRNA template to guide protein synthesis. The more ribosome-protected fragments of specific mRNAs are able to be profiled through a deep sequencing-based tool, the higher level of translation is exhibiting. The increased distribution of ribosome footprints will significantly reprogramming the translation of the targeted mRNA [DOI:10.1038/nrm4069].

The discussion has been supplemented. Please kindly see: Line 285-292 on page 8.

14) The overall logic and rationale for the study should be improved. Why are these modifications chosen and why SOX9 focused on and not the other targets?

Response:

Thanks for your advice on this manuscript. The overall logic and rational for

the study has been improved. The reasons to choose m⁵C and m⁶A modifications and focus on Sox9 as target have been stated in the Introduction and Results sections. Please kindly see: Line 68-70 on page 2, Line 79-84 on page 3, Line 115-116 on page 4, Line 123-124 on page 4.

15) In Figure 2 the images of KD of METTL3 and NSUN4 seem different to the controls but are different between each other. A more careful assessment of what stage of differentiation is blocked or what are these cells?

Response:

①

We applied toluidine blue staining to detect the chondrogenic differentiation. The result revealed that the chondrogenic-induced pellets showed intense metachromasia, but siNsun4 and siMettl3 pellets had no significant change. The data indicates that chondrogenic differentiation is blocked after knockdown Nsun4 or Mettl3. And knockdown Nsun4 or Mettl3 may have different effects on the synthesis rate of downstream target genes Aggrecan and Col2, resulting in the differences between the two groups.

②Several experimental studies have confirmed that Sox9 abundantly existed in cartilage progenitor cells and chondrogenic cells, which is a necessary condition for maintaining the chondrocytes phenotype. Consecutively, Sox9 inhibits the differentiation of chondrocytes into pro-hypertrophic chondrocytes and does not participate in the further differentiation of hypertrophic chondrocytes at the end stage, and then the expression of Sox9 is turned off[doi.org/10.3389/fcell.2021.664168]. Our experimental results showed that Sox9 was highly expressed, indicating an early stage of differentiation.

16) The authors should show in vivo that KD of these factors can block chondrocyte differentiation or effect regeneration not just improve it with overexpression.

Response:

Thanks for the comments. Because of the limited time and conditions, a long-term observation on animals will be further studied.

17) Some form of pathway analysis could be used to strengthen their

conclusions about cell programs and cell types.

Response:

Thanks for your kind advice. The core of this paper is to find that the complex of Nsun4 and Mettl3 promotes chondrogenic differentiation of BMSCs via modulating the translation reprogramming of Sox9. It has reported that other signaling pathways also play roles in chondrogenic differentiation, including Wnt signaling pathway, FGF signaling pathway, Notch signaling pathway and so on. We will explore these signaling pathways in the future study as well.

Reviewers' comments:

Reviewer #1 (Remarks to the Author):

The authors have revised the manuscript according to the suggestion.
The manuscript can be accepted for publication.

Reviewer #2 (Remarks to the Author):

The authors made improvements to the manuscript, but it is very important to notice that the study still lacks key controls, which renders a central claim (i.e., the involvement of the m5C RNA modification in the observed phenotypes) more or less unsubstantiated.

Original point 1: The quality of the methylene blue controls is unacceptable, as the dots are barely detectable. Also, the Bioanalyzer traces provided in the rebuttal letter are by no means sufficient to exclude contamination of the mRNA samples with tRNAs and/or rRNAs. This is a key point that needs to be addressed with standard methodology (high-quality Northern blots and/or shotgun sequencing).

Original point 3: The bisulfite sequencing results are still uninterpretable. What is the "probability" that is provided by the authors? There are accepted standards on how to display bisulfite sequencing results and the authors need to consult the literature and adapt these standards for their own data visualization. In this context, the standard internal controls that establish the site specificity of the methylation mark and discriminate it from artifacts due to incomplete bisulfite deamination need to be provided.

Original point 4: The new results do not appear to be particularly convincing and effect sizes appear rather small. In light of the authors' problems to convincingly demonstrate the presence of m5C in Sox9 mRNA (see my points above), the A2030 and C2062 mutations need to be tested separately. This would also represent an important internal control.

Reviewer #3 (Remarks to the Author):

The authors have responded to the concerns.

Reviewer #2 (Remarks to the Author):

The authors made improvements to the manuscript, but it is very important to notice that the study still lacks key controls, which renders a central claim (i.e., the involvement of the m5C RNA modification in the observed phenotypes) more or less unsubstantiated.

Original point 1: The quality of the methylene blue controls is unacceptable, as the dots are barely detectable. Also, the Bioanalyzer traces provided in the rebuttal letter are by no means sufficient to exclude contamination of the mRNA samples with tRNAs and/or rRNAs. This is a key point that needs to be addressed with standard methodology (high-quality Northern blots and/or shotgun sequencing).

Response:

1) Thanks for your positive comments. Now, we have revised the methylene blue controls. Please kindly see: New Figure 1E-F, Figure 2H-I, Figure 3C-D, Supplementary Figure 2G-H, Supplementary Figure 3G-J.

2)

A

Purified RNA from total RNA			
Total RNA (µg)	Purified RNA (µg)	Recovery (%)	A _{260nm} /A _{280nm}
30	0.6	2	1.9

B

C

Total RNA was isolated using TRIzol reagent, and mRNA was purified using Dynabeads™ mRNA Purification Kit (Invitrogen, 61006). Theoretically, about 80 percent of the total RNA in mammalian cells is rRNA, and mRNA constitutes only about 5 percent of the total RNA[DOI: 10.1038/sdata.2015.63]. The recovery percentage for experiment is approximately 2%, which is consistent with the efficiency captured by oligo dT dynabeads (Fig. A) [DOI: 10.1073/pnas.1232347100]. The purity of the mRNA was analyzed by denatured agarose gel electrophoresis[DOI:10.1016/j.ab.2008.04.016]. The result showed that two sharp bands of rRNA (28S and 18S) were very faint in the gel of purified mRNA (Fig. B). To check the compatibility of isolated mRNA, northern blot was used. The result revealed that the highly purified and intact mRNA was isolated (Fig.C). In conclusion, the enrichment mRNA method is effective enough to get rid of rRNAs from the total RNA and makes rather pure mRNA for performing further studies. The probe sequences are as follows:

Sox9 mRNA: 5'-ccggcgagcactcggggcaatcccagg-3'

GAPDH mRNA: 5'-atcaccatcttcaggagcgagatccctcca-3'

Original point 3: The bisulfite sequencing results are still uninterpretable. What is the “probability” that is provided by the authors? There are accepted standards on how to display bisulfite sequencing results and the authors need to consult the literature and adapt these standards for their own data visualization. In this context, the standard internal controls that establish the site specificity of the methylation mark and discriminate it from artifacts due to incomplete bisulfite deamination need to be provided.

Response:

1)

Thanks for the comments. Now we have changed the “probability” to “percentage of m5C(%)”. Please kindly see: New Fig. 3M.

2)

A

Untreated Results:

```
4329-gggccucacg  auccuucuga  ccuuuugggu  uuuaagcagg  aggugucaga  aaaguuacca
4389-cagggauaac  uggcuugugg  cggccaagcg  uucauagcga  cgucgcuuuu  ugauccuuCg
4449-augucggcuc  uuccuaucau  ugugaagcag  aaucaccaa  gcgugggauu  guucaccac
4509-uauagggaa  cgugagcugg
```

Treated Results:

```
4329-ggguuuuuag  auuuuuuuga  uuuuuugggu  uuuaaguagg  agguguuaga  aaaguuuuu
4389-uagggauaau  ugguuugugg  ugguuagug  uuuauguga  uguuuuuuu  ugauuuuuCg
4449-auguugguuu  uuuuuuuuu  ugugaaguag  auuuuuuuu  uguuuggauu  guuuuuuuu
4509-uauagggaa  ugugaguugg
```

B Conversion Efficiency (C to T): C: 99.5%

1 100

1- GGGT TTTATGAT TTTTTGA TTTTTGGG TTTAAGTAGGAGGTGTAGAAAAAGTTATT ATAGGGATAATTGG TTTGTGGTGGTT AAGTGTTCATAGTGA
 2- GGGT TTTATGAT TTTTTGA TTTTTGGG TTTAAGTAGGAGGTGTAGAAAAAGTTATT ATAGGGATAATTGG TTTGTGGTGGTT AAGTGTTCATAGTGA
 3- GGGT TTTATGAT TTTTTGA TTTTTGGG TTTAAGTAGGAGGTGTAGAAAAAGTTATT ATAGGGATAATTGG TTTGTGGTGGTT AAGTGTTCATAGTGA
 4- GGGT TTTATGAT TTTTTGA TTTTTGGG TTTAAGTAGGAGGTGTAGAAAAAGTTATT ATAGGGATAATTGG TTTGTGGTGGTT AAGTGTTCATAGTGA
 5- GGGT TTTATGAT TTTTTGA TTTTTGGG TTTAAGTAGGAGGTGTAGAAAAAGTTATT ATAGGGATAATTGG TTTGTGGTGGTT AAGTGTTCATAGTGA
 6- GGGT TTTATGAT TTTTTGA TTTTTGGG TTTAAGTAGGAGGTGTAGAAAAAGTTATT ATAGGGATAATTGG TTTGTGGTGGTT AAGTGTTCATAGTGA
 7- GGGT TTTATGAT TTTTTGA TTTTTGGG TTTAAGTAGGAGGTGTAGAAAAAGTTATT ATAGGGATAATTGG TTTGTGGTGGTT AAGTGTTCATAGTGA
 8- GGGT TTTATGAT TTTTTGA TTTTTGGG TTTAAGTAGGAGGTGTAGAAAAAGTTATT ATAGGGATAATTGG TTTGTGGTGGTT AAGTGTTCATAGTGA
 9- GGGT TTTATGAT TTTTTGA TTTTTGGG TTTAAGTAGGAGGTGTAGAAAAAGTTATT ATAGGGATAATTGG **C**TTGTGGTGGTT AAGTGTTCATAGTGA
 10- GGGT TTTATGAT TTTTTGA TTTTTGGG TTTAAGTAGGAGGTGTAGAAAAAGTTATT ATAGGGATAATTGG TTTGTGGTGGTT AAGTGTTCATAGTGA

Orig. - GGG**C**C**T**CACGAT**C**CTT**C**TGAC**C**TTTTGGGTTTTAAG**C**AGGAGGTGT**C**AGAAAAAGTTA**C****C**ACAGGGATA**C**TGG**C**TTGTGG**C**GG**C**AA**C**GT**C**ATAG**C**GA

101 200

1- TGTT GTTTTTGA TTTT**C**GATGTTGGTTTTTT TATTATTGGAAGTAGAA TTTATT AAGTGTGGATTG TTTATTT ATTAATAGGG AATGTGAGTTGG
 2- TGTT GTTTTTGA TTTT**C**GATGTTGGTTTTTT TATTATTGGAAGTAGAA TTTATT AAGTGTGGATTG TTTATTT ATTAATAGGG AATGTGAGTTGG
 3- TGTT GTTTTTGA TTTT**C**GATGTTGGTTTTTT TATTATTGGAAGTAGAA TTTATT AAGTGTGGATTG TTTATTT ATTAATAGGG AATGTGAGTTGG
 4- TGTT GTTTTTGA TTTT**C**GATGTTGGTTTTTT TATTATTGGAAGTAGAA TTTATT AAGTGTGGATTG TTTATTT ATTAATAGGG AATGTGAGTTGG
 5- TGTT GTTTTTGA TTTT**C**GATGTTGGTTTTTT TATTATTGGAAGTAGAA TTTATT AAGTGTGGATTG TTTATTT ATTAATAGGG AATGTGAGTTGG
 6- TGTT GTTTTTGA TTTT**C**GATGTTGGTTTTTT TATTATTGGAAGTAGAA TTTATT AAGTGTGGATTG TTTATTT ATTAATAGGG AATGTGAGTTGG
 7- TGTT GTTTTTGA TTTT**C**GATGTTGGTTTTTT TATTATTGGAAGTAGAA TTTATT AAGTGTGGATTG TTTATTT ATTAATAGGG AATGTGAGTTGG
 8- TGTT GTTTTTGA TTTT**C**GATGTTGGTTTTTT TATTATTGGAAGTAGAA TTTATT AAGTGTGGATTG TTTATTT ATTAATAGGG AATGTGAGTTGG
 9- TGTT GTTTTTGA TTTT**C**GATGTTGGTTTTTT TATTATTGGAAGTAGAA TTTATT AAGTGTGGATTG TTTATTT ATTAATAGGG AATGTGAGTTGG
 10- TGTT GTTTTTGA TTTT**C**GATGTTGGTTTTTT TATTATTGGAAGTAGAA TTTATT AAGTGTGGATTG TTTATTT ATTAATAGGG AA**C**GTGAGTTGG

Orig. - **C**GT**C****C**TTTTTGAT**C**CT**C**CGATG**C**GG**C****T****C**T**C**TAT**C**ATTGTGAAG**C**AGAATT**C****C****C**AA**C****C**GTGGATTGTT**C**AC**C****C**A**C**TAATAGGGAA**C**GTGAG**C**TGG

C

Primers of Bisulfite sequencing		
Normal primer	Forward	GGGGCCTCACGATCCTTCTGACCTTTTGGG
	Reverse	CCAGCTCACGTTCCCTATTAGTGGGTGAAC
Specific primer	Forward	GGGGTTTTAYGATTTTTTTGATTTTTTGGG (Y=C/T)
	Reverse	CCAACTCACRTTCCCTATTAATAAAATAAAC

To assess efficiency of bisulfite conversion treatment, we used the 28S rRNA as positive control, as the C at position 4447 is generally 100% methylated. (DOI:10.3389/fviro.2021.714475). We performed bisulfite treatment of total RNA, followed by RT-PCR and Sanger sequencing of the C4447 encompassing region of the 28S rRNA. The result showed a complete C-T conversion along the fragment suggesting the absence of methylation on these C residues, and no conversion of C4447 residue confirming the methylation status of this specific C residue (Fig. A). To further assess the conversion rate, the RT-PCR sequencing results of 10 clones were obtained. The sequence analysis showed an average conversion rate of 99.5%, suggesting that bisulfite treatment was efficient (Fig. B). The bisulfite primers of 28S rRNA were shown in Fig. C.

Highlighted **C** represents m⁵C, highlighted **C** represents non-converted cytosine. The original, non-converted RNA sequence with non-methylated **C** highlighted is shown below the converted cDNA sequencing results for comparison.

Original point 4: The new results do not appear to be particularly convincing and effect sizes appear rather small. In light of the authors' problems to convincingly demonstrate the presence of m⁵C in Sox9 mRNA (see my points above), the A2030 and C2062 mutations need to be tested separately. This would also represent an important internal control.

Response:

To further confirm the functional relevance of the A2030 and C2062 modifications, we cloned the 300-nucleotide (nt)-long WT or mutant 3'UTR truncation sequence (from 1950 to 2250) into the pmirGLO luciferase (Luc) reporter (Fig. A-D up). The dual-luciferase assay showed that translation efficiency of Sox9 in siNsun4, siMettl3 and siYthdf2 groups were significantly downregulated than that in 293T NC group in WT reporter, and was not changed in mutations of the m⁵C and m⁶A sites (Fig. A-D down).

Please kindly see: Fig. 4H-I, Fig.S4E-F.

Reviewers' comments:

Reviewer #2 (Remarks to the Author):

Numbering after my original comments:

1. There are still no Northern blots to control for contaminations with rRNAs and tRNAs.
3. The results look convincing, but need to be included as an additional supplementary figure using commonly accepted reporting standards for DNA methylation results.
4. The F-Luc/R-Luc ratio varies by a factor of 5 between individual experiments in control cells ("NC" bars in the figure in the rebuttal letter), while the siRNA knockdowns induce a 1.3-fold change with the WT construct (Fig. 4I). This suggests that the experimental setup is not sufficiently robust to generate conclusive results.

Reviewers' comments:

Reviewer #2 (Remarks to the Author):

Numbering after my original comments:

1. There are still no Northern blots to control for contaminations with rRNAs and tRNAs.

Response:

A list of aligned rRNA database

Sample	clean_reads	Mapped_Reads(%)	Unmapped_Reads(%)
Control-1	54073006	3371950 (6.24%)	50701056 (93.76%)
Control-2	49870564	2239378 (4.49%)	47631186 (95.51%)
Induced-1	47265340	2397418 (5.07%)	44867922 (94.93%)
Induced-2	60785522	3295496 (5.42%)	57490026 (94.58)

During the enrichment of mRNA with polyA tail by oligo (dT) magnetic beads, there were still some rRNA and tRNA residues due to the influence of sample and species. In RNA-seq, the same method was used to purify mRNA (Fig. S2B-C). The short read alignment tool bowtie2 was used to align the clean reads to the ribosome database. The results showed that the proportion of aligned reads has reached about 5%, which indicated that there was still a small amount of rRNA contamination (the table above). However, the purity of mRNA was high, which can reach the requirements of experiment.

3. The results look convincing, but need to be included as an additional supplementary figure using commonly accepted reporting standards for DNA methylation results.

Response:

The control results of RNA methylation have been showed as a supplementary figure. Please kindly see: Supplementary Figure 4, New Table 1.

4. The F-Luc/R-Luc ratio varies by a factor of 5 between individual experiments in control cells ("NC" bars in the figure in the rebuttal letter), while the siRNA knockdowns induce a 1.3-fold change with the WT construct (Fig. 4I). This suggests that the experimental setup is not sufficiently robust to generate conclusive results.

Response:

Thanks for your positive comments. In the last rebuttal letter, the results of the mutant groups were not operated on the same batch as the result of the wild type. In order to further confirm our results, we repeated the experiment

at the same time and got similar results. The difference of si-NC group in individual experiments was mainly due to the difference of vector. After site mutation, the F-Luc/R-Luc ratio decreased, which was consistent with the hypothesis.

For the original result, the WT-vector was transfected for 24h followed by analysis with the Dual-Glo Luciferase System. To accurately value the change, we transfected it for 48h from six independent experiment and confirmed that the siRNA knockdowns induce a 2-fold change with significant statistical difference.

Please kindly see: New Figure 4H-I, Supplementary Figure 5E-F.